**Aragonite saturation states and pH in western Norway fjords:**
**seasonal cycles and controlling factors, 2005-2009**

A. M. Omar[1,2], I. Skjelvan[1], S.R. Erga[3], A. Olsen[2]

[1]: Uni Research Climate, Bjerknes Centre for Climate Research, Bergen, Norway.

[2]: Geophysical Institute, University of Bergen, and Bjerknes Centre for Climate Research, Bergen, Norway.

[3]: Department of Biology, University of Bergen, Bergen, Norway.

## Abstract:

The uptake of anthropogenic Carbon Dioxide ($CO_2$) by the ocean leads to a process known as ocean acidification (OA) which lowers the aragonite saturation state ($\Omega_{Ar}$) and pH, and this is poorly documented in coastal environments including fjords due to lack of appropriate observations.

Here we use weekly underway data from Voluntary Observing Ships (VOS) covering the period 2005-2009 combined with data from research cruises to estimate $\Omega_{Ar}$ and pH values in several adjacent western Norwegian fjords, and to evaluate how seawater $CO_2$ chemistry drives their variations in response to physical and biological factors.

The OA parameters in the surface waters of the fjords are characterized by strong seasonal and spatially coherent variations. These changes are governed by the seasonal changes in temperature, salinity, formation and decay of organic matter, and vertical mixing with deeper, carbon-rich coastal water. Annual mean pH and $\Omega_{Ar}$ values were 8.13 and 2.21, respectively. The former varies from minimum values ($\approx$8.05) in late December - early January to maximum values of around 8.2 during early spring (March-April) as a consequence of the phytoplankton spring bloom, which reduces Dissolved Inorganic Carbon (DIC). In the following months, pH decreases in response to warming. This thermodynamic decrease in pH is reinforced by the deepening of the mixed layer, which enables carbon-rich coastal water to reach the surface, and this trend continues until the low winter values are reached again. $\Omega_{Ar}$, on the other hand, reaches its seasonal maximum ($>$2.5) in mid to late summer (July -Sept), when the spring bloom is over and pH is decreasing. The lowest $\Omega_{Ar}$ values ($\approx$1.3-1.6) occur during winter (Jan-Mar), when both pH and Sea Surface Temperature (SST) are low and DIC is highest. Consequently, seasonal $\Omega_{Ar}$ variations align with those of SST and salinity normalized DIC (nDIC).

We demonstrate that underway measurements of fugacity of $CO_2$ in seawater ($fCO_2$) and SST
from VOS lines combined with high frequency observations of the complete carbonate system
at strategically placed fixed stations provide an approach to interpolate OA parameters over
large areas in the fjords of western Norway.

**1. Introduction:**

The continued emissions of carbon dioxide ($CO_2$) (Le Quéré et al., 2015) are of global
concern, not only because they are the main drivers of anthropogenic global warming, but
also because of the changes in the ocean chemistry they cause (Ciais et al., 2013). The
increase in the atmospheric $CO_2$ concentration drives a net ocean $CO_2$ uptake, which leads to
higher proton ($H^+$) concentration i.e. lower pH, lower concentration of carbonate ion ($CO_3^{2-}$)
and, lower saturation state ($\Omega$) for calcium carbonate in seawater. This process is known as
ocean acidification (OA) (e.g. Royal Society, 2005), and it has direct and indirect effects on
biological activity in the ocean (e.g. Gattuso and Hansson, 2011) including reported inhibition
of biogenic calcification by marine organisms which precipitate 0.5–2.0 Gt of carbon as
calcium carbonate ($CaCO_3$) in the global ocean every year (Bach, 2015).
For the open ocean, the rate of OA has been relatively well documented and understood
during the last decade. Observations from time series stations and volunteer observing ships
in different oceanic regions consistently show systematic changes in surface ocean chemistry
that result from OA. Specifically, long-term negative trends of pH and saturation state for
aragonite ($\Omega_{Ar}$) have been observed (e.g. Lauvset et al., 2015; Bates et al., 2014).
For coastal regions observed rates of pH change largely differ from those expected from
oceanic $CO_2$ uptake alone, as variations in other biogeochemical processes, related for
example to changes in nutrient loading and eutrophication, are important as well (Clargo et al.
2015; Provoost et al., 2010; Wootton et al., 2008).
The Norwegian west coast (Fig. 1) is dominated by fjords, narrow and deep estuaries, carved
by glacial processes, with a sill in the mouth where they connect to the coastal North Sea.
Apart from being important recreation areas and marine pathways, these fjords are important
ecosystems and their physics, marine life, and associated environmental pressures have been
relatively well studied (e.g. Matthews and Sands, 1973; Erga and Heimdal, 1984; Asplin et
al., 2013; Brattegard et al., 2011; Stigebrandt, 2012; and references therein).
However, only a few studies on the carbon cycle of Norwegian fjords exist in the literature,
and these are from the high Arctic at Svalbard (Fransson et al., 2014; Omar et al., 2005). This
hampers our understanding of the natural variability and controls of seawater carbonate
chemistry, which is a prerequisite for sound estimates of OA in these ecosystems. Generally,
in the Northern Hemisphere, high latitude coastal regions are thought to be sinks for
atmospheric $CO_2$, while low-latitude regions are thought to be $CO_2$ sources (Borges et al.,
2005; Cai et al., 2006; Chavez et al., 2007; Chen and Borges, 2009). The few existing studies
of Norwegian fjords confirm the above picture; they act as an annual net sink for atmospheric
$CO_2$ (Fransson et al., 2014; Omar et al., 2005).
The carbon cycle of the northern North Sea, to which the western Norway fjords are
connected, has been well studied (Thomas et al., 2004; 2005; 2007; 2008; Bozec et al., 2005;
2006; Omar et al., 2010). However, observation-based OA estimates are still scarce. Recently,
Clargo et al. (2015) observed a rapid pH decrease in the North Sea, but after accounting for
biological processes, they estimated an ocean acidification rate consistent with concurrent
atmospheric and open ocean $CO_2$ increases over the period they studied, 2001-2011.
Filling the knowledge gap on OA (and generally the carbon cycle) in western Norwegian
fjords is important because these areas are spawning grounds for different fish species
(Salvanes and Noreide, 1993; Johannessen et al., 2014), productions sites for pelagic
calcifiers (Berge, 1962; Erga and Heimdal, 1984; Frette et al., 2004), the home for some coral
reefs (e. g. Fosså et al., 2002), and significant food sources due to the aquaculture industry
which operates there. Observations of the carbon cycle dynamics in the fjord system will not
only further our understanding and ability for prediction, but they will also serve as
benchmarks against which future changes are compared.
In this study, we present first estimates of OA parameters in surface waters of several adjacent
western Norway fjords (Fig. 1), based mainly on weekly underway data from Voluntary
Observing Ships (VOS) covering the period 2005-2009. We combine the underway data with
available station data from research cruises to facilitate a complete description of the seawater
$CO_2$ chemistry in accordance with the recommendations of OA core principles by
McLaughlin et al. (2015). We focus on analyses of $\Omega_{Ar}$ and pH values and evaluate their
variations in response to the physical and biological factors: summer warming and
stratification, spring phytoplankton bloom, and deep mixing during fall and winter. First we
present the mean distribution across the different fjords (Korsfjord-Langenuen-

Hardangerfjord) to understand the spatiotemporal patterns, then we collapse all data into a monthly time series to analyze the seasonal controls and resolve any interannual or multiyear temporal patterns.

**1.1 The study area**

The study area covers, from north to south, the interconnected Raunefjord (centered around 60.27°N; 5.17°E), the Korsfjord (centered around 60.17°N; 5.21°E), Langenuen, and southern parts of the Hardangerfjord, which are all situated along the western coast of mainland Norway (Fig. 1). The area stretches over some 60 km, but the main focus here will be on the area from the Korsfjord to the Hardangerfjord from which the vast majority of the data has been acquired.

The bathymetry and hydrographic conditions of the fjords have been described elsewhere (Helle, 1978; Mathew and Sands, 1973; Bakke and Sands, 1977; Erga and Heimdal, 1984; Asplin et al., 2014). In the following only a brief account, based on the above studies, is given.

The Korsfjord is 690 m deep in its main basin and situated about 25 km south of the Norway's second largest city, Bergen. To the west Korsfjord is relatively well connected to the open coastal ocean of the northern North Sea through a 250 m deep sill at Marsteinen. To the north it connects with the Raunefjord through the 100 m deep strait Lerøysundet - between the islands Sotra and Lerøy. At the eastern end the fjord branches into the smaller and shallower fjords Lysefjord and Fanafjord, and to the southwest it connects with the open coast through the Selbjørnsfjord, which has a sill depth of 180 m at Selbjørn. To the south it connects to the Hardangerfjord through the 25 km long and 300 m deep strait Langenuen.

The Hardangerfjord is a 179 km long fjord ranking as the fourth longest fjord in the world. It stretches from the coastal open ocean in the southwest to the mountainous interior of Norway. Our study includes the southern parts of the fjord. This is bounded by the larger islands Stord and Tysnesøya in the north, the Haugaland peninsula in the south, and the smaller islands Fjellbergøya and Halsnøya on the south/east side. This part of the fjord is over 300 m deep in its basin (around 59.76N; 5.55E) and connects with the smaller fjords Ålfjord and Bjoafjord in the south.

In the fjord system run-off from land mixes with salty water originating from the northward flowing Norwegian Coastal Current (NCC) to produce a typically salinity stratified water column with a complex circulation, forced both by external and internal factors. In particular, the coastal winds have a profound influence on the water circulation in western Norwegian

fjords producing episodic renewal of the deep water that follows periods of prolonged
northerly winds (Svendsen, 1981; Erga and Heimdal, 1984).
Besides wind forcing, the hydrography of the fjords is also influenced by winter cooling,
summer warming, snow melt and run-off. The fjords also receive freshwater through the
NCC, which carries water originating from the Baltic Sea and rivers in the southern North Sea
(Skagseth et al., 2011 and references therein). However, snow melt and run-off are the main
local sources for freshwater since the fjord system is generally ice free. Thus, on seasonal
time scales, salinity drives stratification during spring-summer and the water column is more
homogenous during winter. Additionally, Asplin et al. (2014) reported regular episodes of
water exchange between Hardangerfjord and the NCC that homogenized the upper 50 m of
the fjord by mixing with coastal water. During these events the water temperature inside the
Hardangerfjord regularly becomes identical with that of the adjacent coastal North Sea
(Asplin et al., 2014).
Water exchange with the NCC is important for the fjord ecosystems as it supplies nutrients
and oxygen to the area (Aure and Stigebrandt, 1989). In response, primary production is
enhanced in the fjords, which support rich and diverse marine life (Erga and Heimdal, 1984;
Erga, 1989; Salvanes and Noreide, 1993).
Erga and Heimdal (1984) studied the dynamics of the spring bloom in the Korsfjord and
estimated a total primary production of 74 g C m$^{-2}$ during the period February – June. Further,
they reported light regime and water column stability to be dominant controls of the onset of
the bloom. They also pointed out that changes in the alongshore wind component are
important for the bloom dynamics, with persistent northerly winds inducing upwelling of
nutrient-rich coastal water that promotes blooming while the opposite situation follows on
from persistent southerly winds. During calm periods strong stratification develops, which
can ultimately lead to nutrient exhaustion in the upper water column.
The study area, with its adjacent waters, is ecologically and economically important because
it covers spawning grounds for a number of different fish species (Lie et al., 1978;
Johannessen et al., 2014). Additionally, the largest concentration of coral reefs in western
Norway is found in the Langenuen strait (Fosså, 2015). The fjord system also contributes to
the important aquaculture production that, with its annual fish production of >700 tonnes,
ranks Norway within the tenth place worldwide. About one fifth of this is produced in the
Hordaland County where the fjord system studied here is situated (http://www.diercke.com).
**2. Data and methods**

## 2.1 **Weekly underway VOS data**

Weekly underway measurements of fugacity of $CO_2$ in seawater ($fCO_2$) and SST were obtained aboard the containership MS Trans Carrier (operated by Seatrans AS, Norway, www.seatrans.no). During the study period, the ship sailed from Bergen to ports in southwestern Norway on a weekly basis. It passed through several fjords including the Korsfjord and the Hardangerfjord (Fig. 1), then crossed the North Sea mostly along a transect roughly at 5°E longitude to Amsterdam, Netherlands, and then back on the same route (Omar et al., 2010). The measurement method used aboard MS Trans Carrier was described in Omar et al. (2010). The instrument recorded one $fCO_2$ and SST measurement about every three minutes and automatically shut off when the ship approached ports in Bergen (20-30 km from port ≈60.2°N) and Amsterdam, in order to protect the inlet filter from potentially polluted seawater. Between February and December 2006 the VOS line was serviced by a sister ship, MS Norcliff, which was equipped with the same measurement system during that period. The VOS line was in operation in the period September 2005 to September 2009. Data acquired between 59.74°N – 60.16°N and 5.17°E – 5.58°E (The Korsfjord, Langenuen, and southern parts of the Hardangerfjord) are used for the current analyses. This dataset will be referred to as the UW (e.g. UW $fCO_2$ and UW SST) which stands for underway. The UW data from the years 2005, 2006, and 2007 are available from the SOCAT database (http://www.socat.info/), the 2008 and 2009 UW data has been submitted for the SOCAT Version 4 release.

## 2.2 Cruise and fixed station data

We augment the VOS data with station data acquired during scientific cruises in the study area in the period 2007-2010 and in 2015, and during regular visits (1-4 times per month) to a fixed station in the Raunefjord in 2007 and 2008. Table 1 summarizes details of these three datasets, which will be referred to as the CS, 2015 and RF datasets, respectively.

Five of the cruises were conducted in the Korsfjord and the Raunefjord (Fig. 1, Table 1) onboard RV Hans Brattstrøm as part of the EU FP7 educational project CarboSchools (CS) in 2007-2010. The CS dataset covers mainly the spring and summer seasons reflecting the somewhat opportunistic nature of the sampling campaign. The 2015 cruise took place during fall (September 24) as part of the Ocean Acidification project funded by the Norwegian Environment Agency, and measurements were taken at three stations in the Korsfjord, Langenuen and southern Hardangerfjord (Fig.1, red squares).

During each of the above cruises water samples were collected for analyses of parameters
including DIC, total alkalinity (TA), salinity and temperature at 1-2 stations. The DIC
concentrations were determined by the coulometric method (e.g. Johnson et al., 1993) with a
precision of $\pm1$ µmol $kg^{-1}$. TA was measured by potentiometric titration with strong acid
(HCl), and a precision of $\pm2$ µmol $kg^{-1}$. Accuracy was checked by using Certified Reference
Material supplied by A. Dickson (SIO). Once all samples have been corrected with respect to
offsets determined from the CRM measurements, the DIC and TA measurements were
accurate to within the respective measurement precision (above). Only surface data
(depth<=4m) from within the geographical rectangle 59.74-60.34°N and 5.17-5.55°E were
used in the current study.
The Department of Biology, University of Bergen has acquired CTD (SAIV) data from a
fixed station in Raunefjord (RF) during 27 days in 2007 and 35 days in 2008 as part of a
monitoring program close to the Marine Biological field station at Espegrend. These data
contained temperature and salinity profiles with one meter resolution. Averages of the
uppermost five meters have been used in this study and will be referred to as the RF dataset.
**2.3 In situ pH sensor data**
In January 2012 we carried out an evaluation of two pH sensors of the type Submersible
Autonomous Moored Instruments (SAMI_pH, second generation) at the Marine Biological
field Station at the eastern shore of the Raunefjord. The sensors were suspended from a
wooden frame attached to the floating docks around a raft-house in the fjord –some hundred
meters from land. The instruments were submersed at about one meter depth in the fjord and
were left for 50 hours starting 24.01.2012 10:00 GMT, recording one measurement each hour.
A full description of the measurement method for these instruments is found at
http://www.sunburstsensors.com/. In addition to pH, these instruments also recorded the
seawater temperature and they have measurement precision and accuracy of <0.001 and +/-
0.003 pH units, respectively. During the test, salinity was also recorded using a Seaguard
RCM from Aanderaa Data Instruments. These sensor data were used to assess the uncertainty
in our pH values estimated as described in section 3.1.
**2.4 Methods**
2.4.1 Complete seawater $CO_2$ chemistry from SST and $fCO_2$
A complete description of the seawater $CO_2$ chemistry from the UW SST and UW $fCO_2$ data
collected onboard MS Trans Carrier has been obtained through a 3-step procedure. This is
similar to the procedure described in Nondal et al. (2009) with the main modification being
that in the current study, sea surface salinity (SSS) was determined from empirical
relationship with SST.
First, the RF dataset has been used to determine the regional SSS versus SST relationship.
The RF data was chosen for this purpose because it covered all seasons well, both in 2007 and
2008. The identified regional SSS-SST relationship allowed us to estimate a SSS value for
each UW SST observation from MS Trans Carrier.  This step was necessary because the total
number of measured SSS values were less than 150 data points, while the available underway
SST and $fCO_2$ data were much more numerous (> 9900 data points), covering most of the
study area during the years 2005-2009. The remaining SST and SSS data (CS, and from
sensors) were used for evaluation to verify that SST-SSS relationship is valid for the whole
study area (section 3.1). Salinity values estimated from SST will be denoted as SSS(sst).
Second, we determined TA from SSS(sst) and SST using an algorithm we identified for the
region using the CS dataset. This allows us to estimate a corresponding alkalinity value for
each UW $fCO_2$ observation obtained from MS Trans Carrier. Alkalinity values estimated from
measured SSS and SST data will be denoted as TA(sss), whereas TA values estimated from
SSS(sst) and SST values will be denoted as TA(sst).
The UW $fCO_2$ together with TA (sst), UW SST, and SSS(sst) were then used to characterize
the full seawater $CO_2$ chemistry using CO2SYS (Lewis and Wallace, 1998; van Heuven et al.,
2011), with K1 and K2 constants from Lueker et al. (2000). The CO2SYS calculation also
gives DIC, pH, $\Omega_{Ar}$ and all other seawater $CO_2$ chemistry variables. The data estimated using
this three stage procedure will be denoted pH(sst) and $\Omega_{Ar}$(sst) and are the main focus of this
study.
pH and $\Omega_{Ar}$ values based on TA(sss) and $fCO_2$ will be denoted as pH(sss) and $\Omega_{Ar}$(sss),
whereas values that are either measured or computed from measured TA and DIC will be
denoted as simply pH and $\Omega_{Ar}$. nDIC denotes the DIC values normalized to constant salinity
(the mean value) according to Friis et al. (2003) with freshwater end member DIC
concentration of 1039 µmol kg$^{-1}$ inferred from the cruise data. An overview of the symbols
used for estimated and derived quantities used in this study is given in Table 2.

## 3. Results and discussion

### 3.1 Correlations and validations

In this section we present the regression equations identified in this study in addition to validating the various estimation procedures used by comparing the estimated values with those measured/computed. The results of these comparisons are summarized in Table 3. For each comparison, the coefficient of determination ($R^2$) and the significance level (*p*-value) are used as metrics for the goodness of the correlation while the associated root-mean-square error (rmse) is benchmarked against the maximum target uncertainties developed by the Global Ocean Acidification Network (GOA-ON) and the California Current Acidification Network (C-CAN) of ±0.2 for $\Omega_{Ar}$ (McLaughlin et al., 2015), which corresponds to maximum uncertainties of ±0.02, ±1.25 or ±1.8 in pH, SST, or SSS, respectively.

The regional SST-SSS relationship obtained from the RF dataset is given by Eq. 1 and is depicted in Fig. 2a (filled symbols). Despite a clear covariation between SST and SSS, there is a lot of scatter in the data and the statistics of the regression equation is not particularly strong (Eq. 1). The observed correlation most probably arises from the annual cycles; during summer the study area embodies warm water diluted by runoff, whereas during winter the surface water is colder and saltier due to little or no runoff. The magnitude of these annual variations varies with time and space and this is reflected by the high degree of scatter in the relationship. Consequently, the identified regression model is able to explain only 27% of the salinity variations. Nonetheless, the independent station and sensor data (dots, squares, and stars), which have been acquired from the whole study area in different seasons, falls into a pattern around the relationship described by Eq.1 with a root-squared-mean error of 0.81 psu. Thus, from here on we assume that Eq. 1 is able to estimate the seasonal SSS variations across the whole study region. To verify this we have compared the monthly averages of RF_SSS data with values obtained using Eq.1 and monthly RF_SST. As shown in the last row of Table 3, the estimated values were significantly correlated with the monthly RF_SSS ($R^2$=0.65 and p=0.002) and the resulting rmse of 0.3 was lower than the benchmark values of ±1.8.

$$SSS = -0.142 SST + 32.09 \text{, for SSS>29; } R^2\text{=0.27; n=61; rmse=1.2.} \tag{1}$$

As further verification that the RF SST dataset is spatially representative, we compared it with the chronologically co-located UW SST that have been acquired onboard Trans Carrier across

the whole study area. The two datasets were found to be almost identical (Fig. 2b; 3[rd] row
Table 3).
The relationship between TA, SSS and SST is given by Eq. 2 according to:

$TA = 32.09 SSS - 4.39 SST + 1210$, $R^2$=0.90; n=23; rmse=13.0 µmolkg$^{-1}$.          (2)

Alkalinity is a semi-conservative parameter and is normally modelled as linear functions of
salinity (e.g. Millero et al., 1998; Bellerby et al., 2005; Nondal et al, 2009). However, using a
multi-parameter linear regression with SST and SSS as independent parameters improved the
regression statistics considerably ($R^2$=0.90; n=23; rmse=13.0 µmolkg$^{-1}$) compared to a linear
regression with only SSS ($R^2$=0.67; n=23; rmse=24.0 µmolkg$^{-1}$). This is probably because
SST acts as an indicator of the effect of nutrient cycling on TA in agreement to what has been
reported for the open Atlantic Ocean (Lee et al, 2006).
To estimate a corresponding TA value for each UW fCO$_2$ observation obtained from MS
Trans Carrier, we used salinity values estimated from the UW SST data by using Eq.1. The
results (denoted as SSS(sst)) were then inputted into Eq.2 to obtain TA(sst) (see Table 2 for
nomenclature). The fact that TA(sst) are based on SSS(sst) rather than measured SSS values
introduce an additional error in the estimated pH(sst) and $\Omega_{Ar}$(sst). In order to assess this error
we compared pH(sst) and $\Omega_{Ar}$(sst) with values based on the cruise data, i.e. pH(sss) and
$\Omega_{Ar}$(sss). First, we computed pH(sss) and $\Omega_{Ar}$(sss) by combining all available measured SSS,
estimated TA(sss) from Eq. 2, and co-located UW SST and UW fCO$_2$. Then we repeated the
calculations, but this time we replaced the measured SSS with estimated SSS(sst) from Eq. 1
to compute pH(sst) and $\Omega_{Ar}$(sst). The very strong linear relationships between the resulting
values in Figs. 2c and 2d (circles) confirms that the estimated pH(sst) and $\Omega_{Ar}$(sst) reproduce
closely the measurement-based values of pH(sss) and $\Omega_{Ar}$(sss) for the whole study area. This
is also evident from comparison statistics on rows 1 and 2 in Table 3 which show that
measured based values are well correlated ($R^2 \approx 1$, p<0.001) with those estimated with rmse
values of 0.003 and 0.04 for pH(sss) and $\Omega_{Ar}$(sst), respectively, which are well within the
aforementioned maximum target uncertainties developed by the C-CAN (last column in Table

311    3).

To quantify the total error associated with the pH(sst) and $\Omega_{Ar}$(sst) estimates, we considered
two main sources for error. First we computed the residuals (estimated – measurement-based)
using the data shown in Figs. 2c and 2d (including the sensor data). The mean difference for
the whole study area was 0.002 +/- 0.004 and 0.005 +/- 0.08 for pH and $\Omega_{Ar}$, respectively.
Thus, the maximum probable error from this source is 0.006 and 0.09 for pH and $\Omega_{Ar,}$
respectively. Additionally, we estimated that the computed and/or measured pH values
included an error of 0.007 pH units, which under the current conditions (mean TA, $fCO_2$,
SST, and SSS) would give an error of 0.038 in $\Omega_{Ar}$. These two error estimates were combined
(as the square root of sum of squares) to determine the total error in our estimates, which were
found to be ±0.01 and ±0.1 for pH and $\Omega_{Ar}$, respectively. It must be noted that the above total
error was derived from all available observational data including the in situ sensor data
(shown in Fig. 2c and in described section 2.3), which are the only wintertime measurements
used in this study. This is important because the lack of wintertime data in the CS dataset
which was used for the identification of AT-SSS/SST relationship (Eq. 2) means that
wintertime AT(sst) might be overestimated so that corresponding pH(sst) values would be
overestimated. In fact, during the aforementioned comparison between pH(sst) and measured
pH we noted that for this particular dataset pH(sst) overestimated the measurements.
However, the estimates were consistent with the observations to within the total error of ±0.01
pH units. Thus, by utilizing the above total errors, we also accounted for the effect of this
possible caveat of Eq. 2 arising from the lack of wintertime TA measurements.
From the above we conclude that we are able to estimate pH(sst) and $\Omega_{Ar}$(sst) across the
whole study area and during all seasons with a total errors of ±0.01 and ±0.1 for pH and $\Omega_{Ar}$,
respectively.
**3.2 Spatiotemporal variations**
In order to present the mean distributions across the different fjords throughout the annual
cycle, we condensed the data into one virtual year by projecting it onto non-equidistant
rectangular grids using the "weighted-average gridding" method of the Ocean Data View
software (Schlitzer, 2015). As evident from Fig. 3, there is a clear seasonality (for the
interannual changes see section 3.3) in both pH(sst) and $\Omega_{ar}$ (sst). The former varies between
minimum values (8.05) around New Year to typical maximum values of around 8.25, which
occur during the late winter and/or spring (March-April). This increase is due to the reduction
of DIC (Fig. 3d), induced by the phytoplankton spring bloom. This clearly counteracts and
outweighs the negative effect on pH of warming the water column during this period.
However, during April/May, the effect of warming begins to dominate and pH(sst) starts
decreasing. By September the SST starts decreasing, while pH continues to drop. This is due
to the effect of the fall mixing, which enables carbon-rich coastal water to reach the surface
layer, as mentioned in section 1, and is reflected by increasing DIC during this period (Fig.
3d).
The mean distribution of $\Omega_{Ar}$(sst) also shows a significant seasonal variation. There are three
factors that drive this: (i) reduced concentrations of DIC by the spring bloom enriches the
concentration of carbonate ions, (ii) $\Omega_{Ar}$(sst) increases with rising temperature so that
warming during the summer actually reinforces the increase of $\Omega_{Ar}$ initiated by biological
carbon uptake, and (iii) reduced TA due to freshwater input from runoff and mixing of deeper
carbon-rich water into surface layer reduce $\Omega_{Ar}$(sst) during fall. Thus, $\Omega_{Ar}$(sst) reaches its
maximum (>2.5) in July-September, when the spring bloom is over and pH has already
started decreasing (Fig. 3a, c). The lowest $\Omega_{Ar}$(sst) values (≈1.3-1.6) occur during winter
(January-March) when both pH and SST are low, despite TA is high due to high SSS values.
The decoupling in the seasonal cycles of pH and $\Omega_{Ar}$ clearly supports the case that pH alone is
not an adequate measure of ocean acidification, in accordance with the C-CAN
recommendation that "measurements should facilitate determination of $\Omega_{Ar}$ and a complete
description of the carbonate system, including pH and $pCO_2$" (McLaughlin et al., 2015).
The above described seasonal variations in pH(sst) and $\Omega_{Ar}$(sst) are spatially more or less
coherent within the whole study area, except for the slight south-north gradient during May-
September, with highest values south of 60°N (see Fig. 3a,c). All in all, during summertime
the study area embodied warm surface water with high $\Omega_{Ar}$(sst) and intermediate pH(sst)
values. During winter, the surface water is cold with low $\Omega_{Ar}$(sst) and pH(sst) values.
**3.3 Controls of seasonal variability and trends**
To investigate the seasonal variability more thoroughly, we computed monthly averages of
pH(sst), SST, $\Omega_{Ar}$ (sst ), and nDIC(sst) for one composite year. Then we quantified the effect
of DIC, TA, SST and SSS on the monthly changes of pH(sst) and $\Omega_{Ar}$ (sst ) in order to gain
more insight into the processes governing the seasonal variations and their relative importance
(Fig. 4).
For pH(sst) we used the decomposition method described in Lauvset et al. (2015) to quantify
the importance of different parameters. This method estimates the monthly pH changes
expected from corresponding changes observed in SST, SSS, DIC, and TA as well as their
sum. The results are shown on Fig. 4 (a-e) where it can be seen that DIC is the most important
driver followed by SST and TA, whereas SSS had a negligible effect (not shown) on the
seasonal pH variations. We also note that the effects of SST and TA combined are nearly
equal to, but opposite to that of DIC (Fig. 4c,d,e). As a result, the sum of all effects is <0.06
pH units, and compares well to the observed amplitudes (Fig. 4a), meaning that the
decomposition model is able to account for the observed seasonal changes. Note also the TA
control is identical to that of SST (Fig. 4c,e). The reason for this is that TA values used here
are obtained from SSS(sst) and SST using Eq. 2, which in effect means that they are based on
SST. This emphasizes the need for measured SSS and TA values when the objective is to
analyze the controls of pH and $\Omega_{Ar}$(sst) variations.
For $\Omega_{Ar}$(sst) we investigated the importance of different controls (DIC, TA, SST, SSS) by
varying them independently over their observed range, while holding all other drivers
constant,  and re-computing $\Omega_{Ar}$(sst). The magnitude of the standard deviation of the results is
indicative of the importance of the varying drivers. The result of this exercise is shown on
Fig. 4f-i. Evidently, the variations of SST and SSS are the least important drivers for $\Omega_{Ar}$(sst)
seasonal changes, since varying these parameters induces changes that are about an order of
magnitude less than the observed seasonal amplitude in $\Omega_{Ar}$(sst). On the other hand, changing
DIC and TA (Fig. 4h,i) induces changes that are comparable to the seasonal amplitude
observed in $\Omega_{Ar}$(sst) (Fig. 4a). We therefore conclude that seasonal changes in DIC and TA
are the most important driver for changes in $\Omega_{Ar}$(sst).
From the above we conclude that the main drivers of $\Omega_{Ar}$(sst) are DIC and TA, whereas for
pH(sst), SST also has a significant impact. This means that the formation and destruction of
organic matter together with upwelling of carbon-rich coastal water, seasonal warming and
cooling, and runoff inputs, are the processes that govern most of the seasonal variability of
OA parameters within the study area. It then follows that interannual variability in the above
processes would lead to corresponding variations in pH(sst) and $\Omega_{Ar}$(sst). Such interannual
changes are evident from the monthly time series (Fig. S1), where the rate of seasonal change
differs between the years, both for SST and DIC normalized to the mean salinity (nDIC)
according to Friis et al. (2003). Additionally, for SST, the extreme values also change
between years. These changes are in turn reflected in the pH(sst) and $\Omega_{Ar}$(sst) for which the
amplitude of the interannual variability (IAV), calculated as the temporal standard deviation,
is presented in Table 4. For pH, IAV was normally much lower than the seasonal changes and
ranged between 0.01 and 0.02 although higher changes were observed during the months
April (0.04), and July and October (0.03). Similarly, for $\Omega_{Ar}$(sst), the IAV was typically 0.1
which is much lower than the seasonal changes (section 3.2). Higher IAVs were observed for
June (0.2) whereas November and December showed the lowest IAVs (<0.05).
Quantitatively, the above IAVs are probably lower limits due to the use of constant empirical
relationships for the estimation of SSS and AT (Eqs. 1 and 2). That is, there may be
interannual changes in the relationship between SST and SSS (Eq. 1) and/or between
SSS/SST and AT (Eq. 2). Thus, the use of a constant relationship over the years may have led
to underestimation of the resulting IAV. Consequently, a comprehensive analysis of the
drivers of the IAV was not carried in this study. However, sensitivity computations we
performed showed that year-to-year differences in pH were related to those in $fCO_2$ rather
than SST changes, whereas year-to-year differences in $\Omega_{Ar}$(sst) were more related to those in
SST than $fCO_2$. In any case, the observed year-to-year differences were not systematic, and no
multiyear temporal trend was apparent from the 4-year time series analyzed in this study.

**3.4 Inference of OA parameters from VOS underway data**

Changes in the oceanic $CO_2$-system variables are related through ratios called Buffer Factors.
Specifically, changes in $\Omega_{Ar}$ and pH in response to $CO_2$ variations can be quantified by partial
derivatives ($\gamma_{DIC}$, $\beta_{DIC}$, and $\omega_{DIC}$), which have been defined by Egleston et al. (2010, their
table 1), and the slope of these relationships can be expressed mathematically by:

$$\partial \ln \Omega / \partial \ln CO2 = \gamma_{DIC} / \omega_{DIC} = \frac{DIC - Alk_C^2 / S}{DIC - Alk_C P / HCO_3^-} \tag{3}$$

$$\partial \ln H^+ / \partial \ln CO2 = \gamma_{DIC} / \beta_{DIC} = \frac{(DIC - Alk_C^2)/S}{(DICS - Alk_C^2)/Alk_C} \tag{4}$$

where expressions for the carbonate alkalinity $Alk_C$ and the parameters $P$ and $S$ are defined in
Egleston et al. (2010). We have evaluated the right hand sides of Eqs. 3 and 4, using the CS
cruise data, and the results showed that these quantities change only a few per cents (1.3 and
3.4 %, respectively) due to seasonal changes in the various variables. The ratio $\gamma_{DIC} / \omega_{DIC}$
changed by 1-6 % and ranged from -1.08 to -0.980, while $\gamma_{DIC} / \beta_{DIC}$ changed by 0.5-3 % and
ranged from 0.84 to 0.88. This, together with the fact that equations 3 and 4 can be defined
in terms of $ln(fCO_2)$ instead of $ln(CO2)$ (Egleston et al., 2010; Takahashi et al., 1993),
suggests that in situations where underway surface $fCO_2$ and SST are frequently measured,
while the $CO_2$ system is fully determined only occasionally, an easy way of interpolating the
seasonality in pH and $\Omega_{Ar}$, is to predict them from $fCO_2$. We have implemented this
alternative way of estimating pH and $\Omega_{Ar}$ using the CS cruise data. For the estimation of $\Omega_{Ar}$
we used $fCO_{2@meanSST}$, which is $fCO_2$ adjusted to constant temperature (i.e. at mean SST),
because these normalization improved the regression significantly. Since we were interested
in pH and $\Omega_{Ar}$ we plotted these parameters directly against $ln(fCO_2)$ or $ln(fCO_{2t@meanSST})$. The
results are shown in Fig. 5 and conform to tight relationships between computed pH and
$ln(fCO_2)$ values (Fig. 5a), and between computed $\Omega_{Ar}$ and $ln(fCO_{2t@meanSST})$ (Fig. 5b). Further,
by using linear curve fitting we determined the relationships according to:
$$pH = -0.389\ln fCO_2 + 10.354\,, \text{R}^2\text{=0.99; n=28; rmse=0.005.} \qquad (5)$$
$$\Omega_{Ar} = \exp(-0.6741\ln fCO_{2\,\text{at}\,meanSST} + 4.6422)\,, \text{R}^2\text{=0.94; n=28; rmse=0.07.} \qquad (6).$$
The magnitude of the residuals (computed – estimated) associated with pH and $\Omega_{Ar}$ values
obtained from the above relationships were 0.000 +/- 0.005 and 0.01 +/- 0.06, respectively,
which is comparable to the residuals associated with pH(sst) and $\Omega_{Ar}$(sst) (Table 3). An
advantage of this procedure, however, is that it utilizes much tighter empirical relationships,
involves fewer computational steps, and is based on UW data, which are much more
numerous than station data from oceanography cruises. Thus, it minimizes errors introduced
by intermediate results such as the TA-SSS/SST regression in Eq. 2 and/or seasonal data
coverage. Furthermore, a direct comparison revealed that values obtained from Eqs. 5 and 6
were almost identical to those of pH(sst) and $\Omega_{Ar}$(sst) (Fig. S2) with values for $\text{R}^2$, p-value,
and rmse of 1, 0, and 0.003 for pH; and 1, 0, and 0.02 for $\Omega_{Ar}$. However, it is important to
realize that for the above procedure too, a representative full description of the carbonate
system is necessary for up-to-date determinations of Eqs. 5 and 6. Further, this calibration
data ideally should include high frequency time series observations, since the slopes (i.e. Eqs.
3 and 4) change slightly with the carbonate system variables (e.g. DIC and TA, see Eqs. 3 and
4), which vary on multiple time scales (hours-days-years).  Furthermore, the procedure is
based on measurements of only one of the four master parameters constituting the carbonate
system (i.e. $fCO_2$). Therefore, it only provides a way to interpolate pH and $\Omega_{Ar}$ values, but
cannot support the analyses of controls that have been provided in the proceeding section.
From Fig. 5b we note that lowest $\Omega_{Ar}$ values are associated with the highest $fCO_{2@meanSST}$
values, which occur during late fall and winter. Monitoring of these extreme values are of
special interest because: (i) during late fall and early winter the upwelling of carbon-rich
water occurs and surface water also reflects the properties of the deeper water, and (ii) the rate
of change at this point (lowest $\Omega_{Ar}$, highest $fCO_{2@meanSST}$) indicates the time when under-
saturation of calcium carbonate can be expected in these waters. To estimate this for the
current data we used Eq. 5 and the observation that the slope (i.e. Eq. 3) and intercept
decreased by about 0.0008 and 0.004 for every 1 µatm increase in mean $fCO_{2@meanSST.}$ We
also took into account an uncertainty of ±0.2 in the $\Omega_{Ar}$ estimates and found that $\Omega_{Ar}$ becomes
undersaturated (<1) when mean annual $fCO_{2@meanSST}$ is about 310±70 µatm higher than its
present value (310 µatm). For business as usual emission scenario (RCP 8.5), this is
equivalent to about year 2070±10 if we assume that the development in the ocean follows that
of the atmosphere (i.e. constant disequilibrium between ocean and atmosphere).

## 4. Summary and concluding remarks

On the basis of four years of weekly underway $fCO_2$ and SST data combined with sporadic
data from research cruises, the ocean acidification parameters pH(sst) and $\Omega_{Ar}$(sst) have been
estimated and analyzed for western Norway fjords stretching over more than 60 km from the
Korsfjord, through Langenuen strait, to southern parts of the Hardangerfjord. The total errors
associated with the estimated values, ±0.01 and ±0.1 for pH and $\Omega_{Ar}$, were about 50% lower
than the maximum target uncertainties developed by the Global Ocean Acidification Network.
Strong seasonal variations, more or less spatially coherent over the whole study area, were
found for OA parameters in the surface waters of the fjords. These changes were governed
mainly by the formation and decay of organic matter, vertical mixing with deeper carbon rich
coastal water, and the seasonal changes in SST and SSS. The annual mean pH was 8.13, and
this parameter varies typically between minimum values ($\approx$8.05) around January to maximum
values of around 8.2, which occur during the spring and/or late winter (March-April) as a
consequence of the phytoplankton spring bloom that reduces DIC. However, sometime during
April/May, the effect of warming starts to dominate, and pH(sst) starts decreasing. Later
during fall, deepening of the mixed layer allows carbon-rich coastal water mix into the
surface, which reduces pH until the low winter values are reached again.
The mean value of $\Omega_{Ar}$(sst) was found to be 2.21, and it reached its maximum (>2.5) in mid to
late summer (July to September), when the spring bloom is over and pH has started to
decrease. The lowest $\Omega_{Ar}$(sst) values ($\approx$1.3-1.6), on the other hand,  occurred during winter
(January-March), when both pH and SST are low, and DIC is at its highest.
Strong correlations of pH and $\Omega_{Ar}$ with $fCO_2$ and $fCO_{2@meanSST}$ ($fCO_2$ adjusted to the mean
temperature), respectively provide an approach to interpolate pH and $\Omega_{Ar}$ over large areas in
the fjords of western Norway where underway measurements of $fCO_2$, SST, and SSS are
available. However, both the slopes and the intercepts of these correlations vary slightly with
DIC and TA. Therefore, the most accurate interpolations will be achieved if the relationships
are calibrated with high frequency observations of the complete carbonate system, measured
at few strategically placed fixed stations.
The $\Omega_{Ar}$ - $fCO_{2@meanSST}$ relationship, and the rate of change of its slope and intercept with
DIC, have been used to project the time when under-saturation of calcium carbonate could be
expected to occur in the study area. This is expected to occur in the year 2070, if we assume
business as usual emission scenario (RCP 8.5), and that oceanic $CO_2$ concentrations follow
that of the atmosphere (i.e. constant disequilibrium between ocean and atmosphere).

## 5. Acknowledgements

We are grateful for financial support by the Research Council of Norway (RCN) through the
project FME SUCCESS, and by the Norwegian Environment Agency through the project
*Havforsuring*. The data collection has been financed by the EU IP CARBOOCEAN (Contract
no. 511176-2). This work would not have been possible without the generosity and help of the
liner company SeaTrans AS and the captains and crew of MS Trans Carrier. We are grateful
for the technical assistance provided by Tor de Lange, Kristin Jackson and Tomas Sørlie, and
for encouraging and constructive comments from the reviewers (M. Ribas Ribas and E. M.
Jones) that have improved the manuscript.

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

**Tables:**



**Table 1:** Details of the CarboSchools (CS) and Raunefjord (RF) cruise datasets. The plus sign denotes the parameters for which sampling/measurement were carried out. For the RF dataset, each data point represents the average of five measurements acquired in the upper five meters.

| Datasett; area | Date(m/d/y) | Lon (E) | Lat (N) | Depth (m) | DIC | TA | SST | SSS | Reference/ originator |
|---|---|---|---|---|---|---|---|---|---|
| CarboSchools (CS); Korsfjord/Raunefjord | 04/13/2007 | 5.19 | 60.34 | 1 | + | + | + | + | I. Skjelvan |
| | 04/13/2007 | 5.18 | 60.17 | 1 | + | + | + | + | |
| | 04/25/2007 | 5.19 | 60.34 | 1 | + | + | + | + | |
| | 04/25/2007 | 5.17 | 60.17 | 1 | + | + | + | + | |
| | 09/04/2008 | 5.18 | 60.33 | 1 | + | + | + | + | |
| | 03/12/2009 | 5.17 | 60.15 | 1 | + | + | + | + | |
| | 03/12/2009 | 5.18 | 60.32 | 1 | + | + | + | + | |
| | 03/12/2009 | 5.17 | 60.17 | 1 | + | + | + | + | |
| | 03/12/2009 | 5.18 | 60.33 | 1 | + | + | + | + | |
| | 08/25/2009 | 5.17 | 60.17 | 1 | + | + | + | + | |
| | 08/24/2009 | 5.18 | 60.16 | 1 | + | + | + | + | |
| | 08/24/2009 | 5.18 | 60.16 | 1 | + | + | + | + | |
| | 08/24/2009 | 5.19 | 60.34 | 1 | + | + | + | + | |
| | 08/25/2009 | 5.19 | 60.34 | 1 | + | + | + | + | |
| | 08/25/2009 | 5.2 | 60.34 | 1 | + | + | + | + | |
| | 08/25/2009 | 5.19 | 60.33 | 1 | + | + | + | + | |
| | 08/27/2009 | 5.19 | 60.33 | 1 | + | + | + | + | |
| | 08/27/2009 | 5.19 | 60.33 | 1 | + | + | + | + | |
| | 08/27/2009 | 5.18 | 60.17 | 1 | + | + | + | + | |
| | 08/27/2009 | 5.18 | 60.17 | 1 | + | + | + | + | |
| | 08/27/2009 | 5.18 | 60.17 | 1 | + | + | + | + | |
| | 08/27/2009 | 5.2 | 60.33 | 1 | + | + | + | + | |
| | 09/08/2010 | 5.2 | 60.33 | 1 | + | + | + | + | |
| 2015; Korsfjord | 09/29/2015 | | | 5 | + | + | + | + | I. Skjelvan / A. Omar |

| Datasett; area | Date(m/d/y) | | | | | | | | I. Skjelvan / A. Omar |
|---|---|---|---|---|---|---|---|---|---|
| 2015; Langenuen | 09/29/2015 | | | 5 | + | + | + | + | I. Skjelvan / A. Omar |
| 2015; Hardangerfjord | 09/29/2015 | | | 5 | + | + | + | + | I. Skjelvan / A. Omar |



Table 1(continued)

| Datasett; area | Date(m/d/y) | Lon_E | Lat_N | Depth (m) | DIC | TA | SST | SSS | Reference/ originator |
|---|---|---|---|---|---|---|---|---|---|
| RF; Raunefjord | 01/03/2007 | | | 1-5 | | | + | + | S. R. Erga / J. Egge |
| | 01/23/2007 | | | 1-5 | | | + | + | |
| | 02/13/2007 | | | 1-5 | | | + | + | |
| | 02/27/2007 | | | 1-5 | | | + | + | |
| | 03/07/2007 | | | 1-5 | | | + | + | |
| | 03/13/2007 | | | 1-5 | | | + | + | |
| | 03/27/2007 | | | 1-5 | | | + | + | |
| | 04/10/2007 | | | 1-5 | | | + | + | |
| | 04/17/2007 | | | 1-5 | | | + | + | |
| | 04/23/2007 | | | 1-5 | | | + | + | |
| | 05/08/2007 | | | 1-5 | | | + | + | |
| | 05/19/2007 | | | 1-5 | | | + | + | |
| | 06/05/2007 | | | 1-5 | | | + | + | |
| | 06/12/2007 | | | 1-5 | | | + | + | |
| | 06/19/2007 | | | 1-5 | | | + | + | |
| | 08/31/2007 | | | 1-5 | | | + | + | |
| | 09/04/2007 | | | 1-5 | | | + | + | |
| | 09/11/2007 | | | 1-5 | | | + | + | |
| | 09/18/2007 | | | 1-5 | | | + | + | |
| | 09/26/2007 | | | 1-5 | | | + | + | |
| | 10/02/2007 | | | 1-5 | | | + | + | |
| | 10/09/2007 | | | 1-5 | | | + | + | |
| | 10/18/2007 | | | 1-5 | | | + | + | |
| | 10/31/2007 | | | 1-5 | | | + | + | |
| | 11/27/2007 | | | 1-5 | | | + | + | |
| | 12/11/2007 | | | 1-5 | | | + | + | |
| | 01/02/2008 | | | 1-5 | | | + | + | |
| | 02/05/2008 | | | 1-5 | | | + | + | |
| | 02/21/2008 | | | 1-5 | | | + | + | |
| | 03/05/2008 | | | 1-5 | | | + | + | |
| | 03/11/2008 | | | 1-5 | | | + | + | |
| | 03/25/2008 | | | 1-5 | | | + | + | |
| | 03/31/2008 | | | 1-5 | | | + | + | |
| | 04/08/2008 | | | 1-5 | | | + | + | |
| | 04/22/2008 | | | 1-5 | | | + | + | |
| | 04/29/2008 | | | 1-5 | | | + | + | |
| | 05/06/2008 | | | 1-5 | | | + | + | |
| | 05/13/2008 | | | 1-5 | | | + | + | |
| | 05/20/2008 | | | 1-5 | | | + | + | |
| | 05/27/2008 | | | 1-5 | | | + | + | |

| | | | |
|---|---|---|---|
| 06/04/2008 | 1-5 | + | + |
| 06/11/2008 | 1-5 | + | + |
| 06/17/2008 | 1-5 | + | + |
| 06/24/2008 | 1-5 | + | + |
| 07/01/2008 | 1-5 | + | + |
| 07/08/2008 | 1-5 | + | + |
| 07/16/2008 | 1-5 | + | + |
| 08/12/2008 | 1-5 | + | + |
| 08/19/2008 | 1-5 | + | + |
| 08/26/2008 | 1-5 | + | + |
| 09/02/2008 | 1-5 | + | + |
| 09/09/2008 | 1-5 | + | + |
| 09/16/2008 | 1-5 | + | + |
| 09/23/2008 | 1-5 | + | + |
| 09/30/2008 | 1-5 | + | + |
| 10/07/2008 | 1-5 | + | + |
| 10/14/2008 | 1-5 | + | + |
| 10/21/2008 | 1-5 | + | + |
| 11/04/2008 | 1-5 | + | + |
| 11/20/2008 | 1-5 | + | + |
| 12/19/2008 | 1-5 | + | + |


**Table 2:** Overview of the symbols used for quantities estimated and/or derived from the measurement-based
variables SSS, SST, TA, pH, DIC, $fCO_2$, and $\Omega_{Ar}$.

| Symbol | Meaning |
|---|---|
| TA(sss) | TA values estimated from measured SSS and SST using Eq. 2. |
| pH(sss), $\Omega_{Ar}$(sss) | pH, and $\Omega_{Ar}$ values estimated by combining TA (sss) and $fCO_2$. |
| SSS(sst) | SSS values estimated from SST using Eq. 1. |
| TA(sst) | TA values determined from estimated SSS(sst) and SST using Eq. 2. |
| pH(sst), $\Omega_{Ar}$(sst), DIC(sst) | Values of pH, $\Omega_{Ar}$ and DIC that have been obtained by combining TA (sst) , $fCO_2$ and ancillary variables. |
| $fCO_{2t}$ | $fCO_2$ at the mean temperature |
| $fCO_{2ts}$ | $fCO_2$ at the mean temperature and salinity |
| nDIC | DIC normalized to the mean salinity |


**Table 3:** Results of the comparisons between measured-based and estimated values for pH, $\Omega_{Ar}$, SST (°C), and
SSS. For the first three parameters, the statistics of the linear relationships depicted on Fig 2b-2d are listed. For
SSS, monthly averaged data are compared to estimates obtained with Eq.1 using monthly SST. For SST, the
comparison is carried out to verify that measurements from Raunefjord are representative for the whole study
area (i.e. UW_SST can be estimated by RF_SST), which is implicitly assumed by the use of Eq.1. $R^2$ is the
coefficient of determination, and "rmse" denotes the root-square-mean error. The latter is compared against
benchmarks derived from maximum target uncertainties (Max. uncertainty) developed by Ocean Acidification
Networks (section 3.1). The *p-value* is the probability of no linear relation between the estimated and
measurement-based values.

| Compared variables | Comparison statistics | | | Benchmarks | |
|---|---|---|---|---|---|
| | $R^2$ | p-value | # points | rmse | Max. uncertainty |
| pH_meas/comp. and pH(sst) | 1.00 | <0.001 | 106 | 0.003 | ±0.02 |
| $\Omega_{Ar}$_computed and $\Omega_{Ar}$(sst) | 0.98 | <0.001 | 106 | 0.04 | ±0.2 |
| UW_SST and RF_SST | 0.95 | <0.001 | 61 | 0.49 | ±1.25 |
| SSS and SSS(sst) | 0.65 | 0.002 | 12 | 0.3 | ±1.8 |


**Table 4:** Monthly mean values for pH(sst) and $\Omega_{Ar}$(sst) and associated interannual variability (IAV), computed
as standard deviations, in the study area for the period 2005-2009.

| | | Jan | Feb | Mar | Apr | May | Jun | Jul | Aug | Sep | Oct | Nov | Dec |
|---|---|---|---|---|---|---|---|---|---|---|---|---|---|
| **pH(sst)** | Mean | 8.08 | 8.10 | 8.16 | 8.19 | 8.18 | 8.15 | 8.15 | 8.17 | 8.14 | 8.11 | 8.10 | 8.08 |
| | IAV | <0.01 | 0.01 | 0.04 | 0.01 | 0.02 | 0.02 | 0.03 | 0.02 | 0.02 | 0.03 | 0.02 | 0.02 |
| **$\Omega_{Ar}$(sst)** | Mean | 1.7 | 1.7 | 1.9 | 2.1 | 2.3 | 2.4 | 2.6 | 2.7 | 2.4 | 2.2 | 1.9 | 1.8 |
| | IAV | 0.1 | 0.1 | 0.1 | 0.1 | 0.1 | 0.2 | 0.1 | 0.1 | 0.1 | 0.1 | <0.05 | <0.05 |


**Figure texts:**
**Fig. 1:** An overview map of western Norway with a detailed map of the study area showing the positions from
where cruise and underway data have been acquired. The thick grey arrow indicates the approximate position of
the Norwegian Coastal Current (NCC).
**Fig. 2: A)** RF SSS as a function of SST (filled symbols) with the regression line described by Eq.1. Sampling
month is indicated by the color of the data points. The CS (dots), 2015 (squares), and sensor (stars) datasets are
also shown for comparison with the regression line. **B)** Compares RF SST with chronologically co-located UW
SST acquired from the whole study area during 2008 (blue) and 2007 (red). **C)** Compares pH(sst) with pH values
that have been measured or computed from TA and DIC. Symbols are as in Fig. 1. **D)** Compares $\Omega_{Ar}$(sst) with
$\Omega_{Ar}$ values that have been computed from measured TA and DIC or from measured pH and UW $fCO_2$. Symbols
are as in Fig. 1.
**Fig. 3:** A) Estimated pH(sst), B) UW SST, C) estimated $\Omega_{Ar}$(sst), and D) estimated DIC which have been
normalized to the mean salinity of 30.5 as a function of latitude and time of the year. All data from 2005-2009
have been condensed into one virtual year to resolve the spatial and seasonal variations.
**Fig. 4: left panel:** Monthly pH changes (ΔpH) as observed (**A**) and expected due to: sum of all derivers (**B**), SST
changes (**C**), DIC changes (**D**) and by TA changes (**E**). **right panel:** Standard deviations in monthly mean $\Omega_{Ar}$ as
a result of variations in all parameters (**F**) or only in SST (**G**) in DIC (H) in TA (**I**).
**Fig. 5:** A) and B) pH and $\Omega_{Ar}$ from CS (dots) and 2015 (red squares) cruises plotted as a function of $ln(fCO_2)$
and $ln(fCO_{2@meanSST})$, respectively.
**FigS1:** Time series of monthly variations in **A)** pH(sst), **B)** SST, **C)** $\Omega_{Ar}$(sst) and **D)** nDIC for the whole study
area in 2005-2009.
**FigS2: A)** compares pH(sst) with pH values obtained from Eq. 5 (y-axis). **B)** Compares $\Omega_{Ar}$(sst) with values
obtained from Eq. 6 (y-axis).


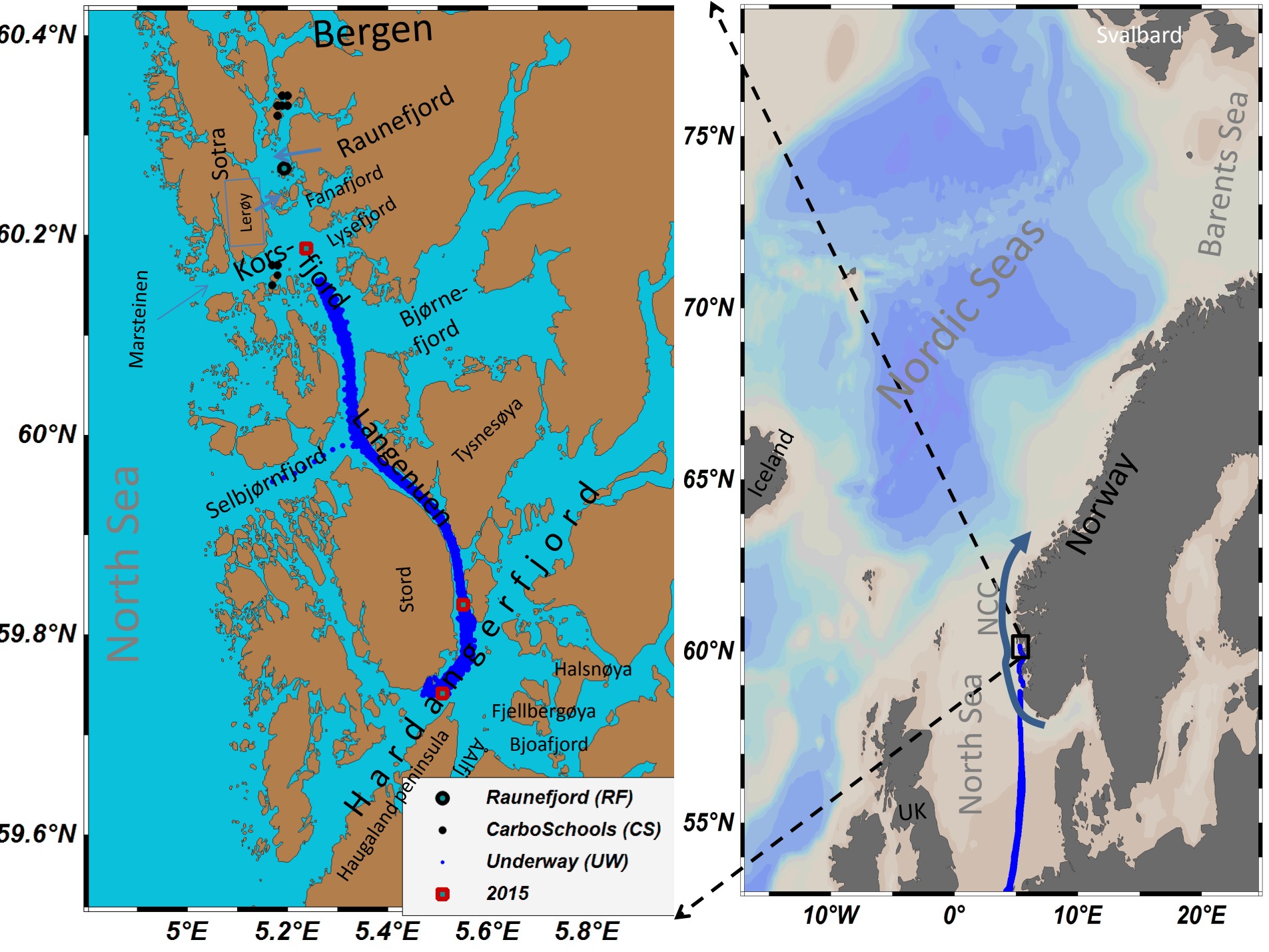

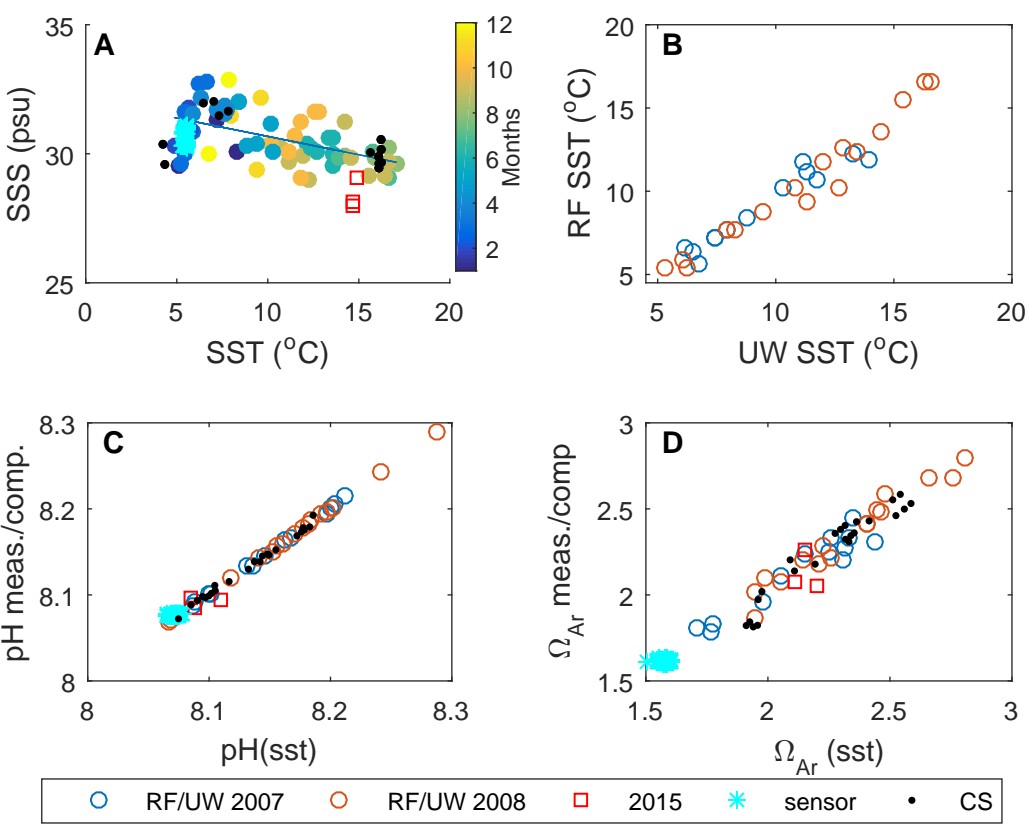

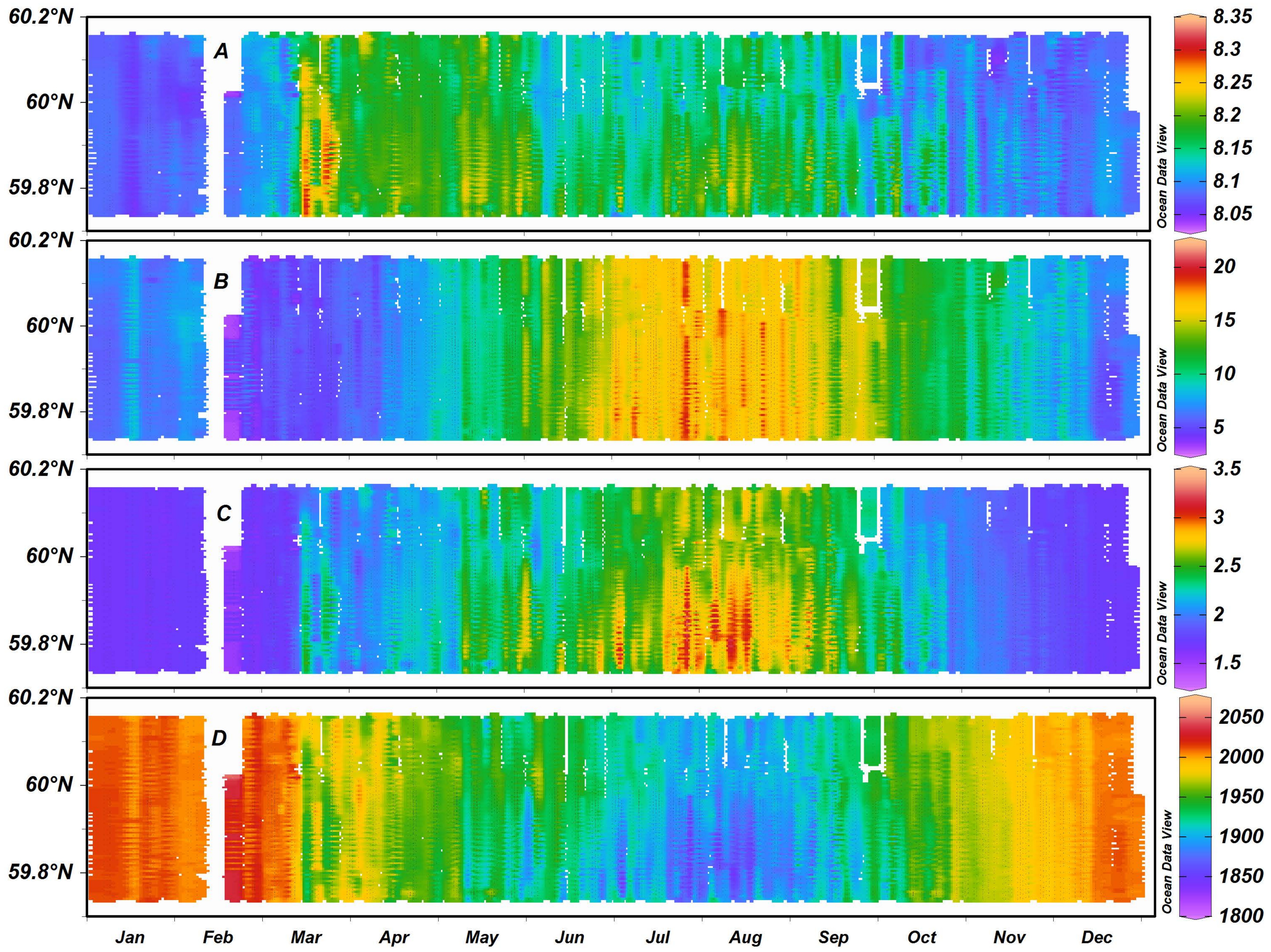

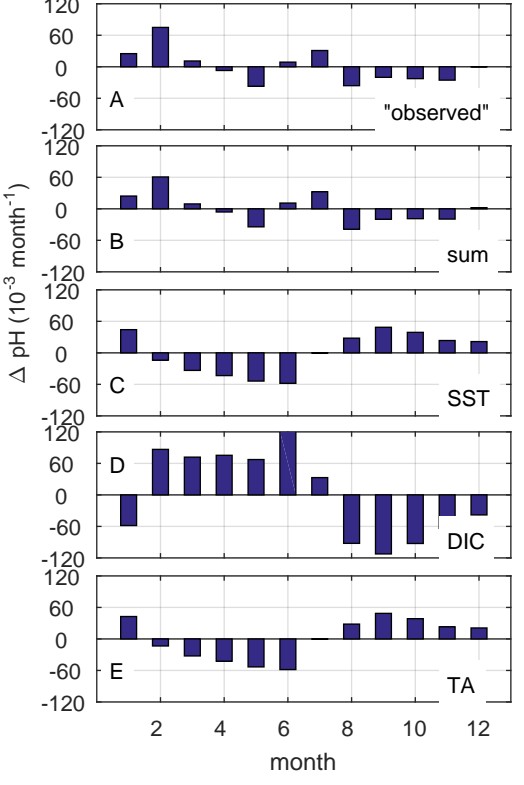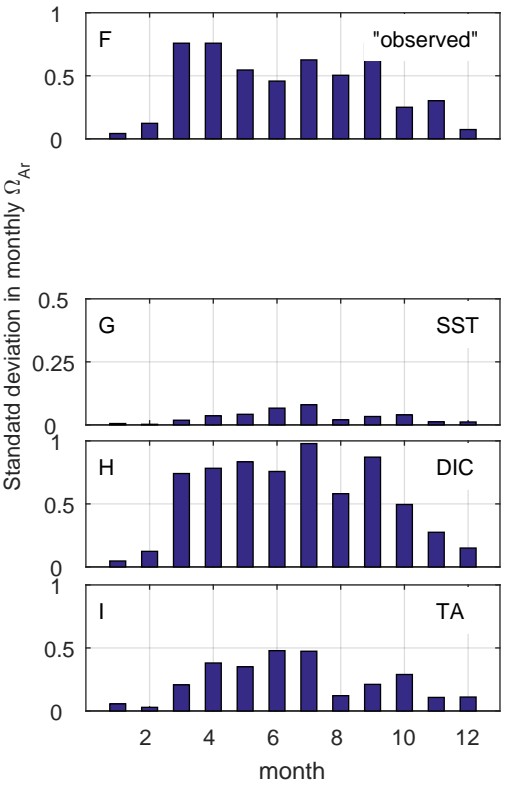

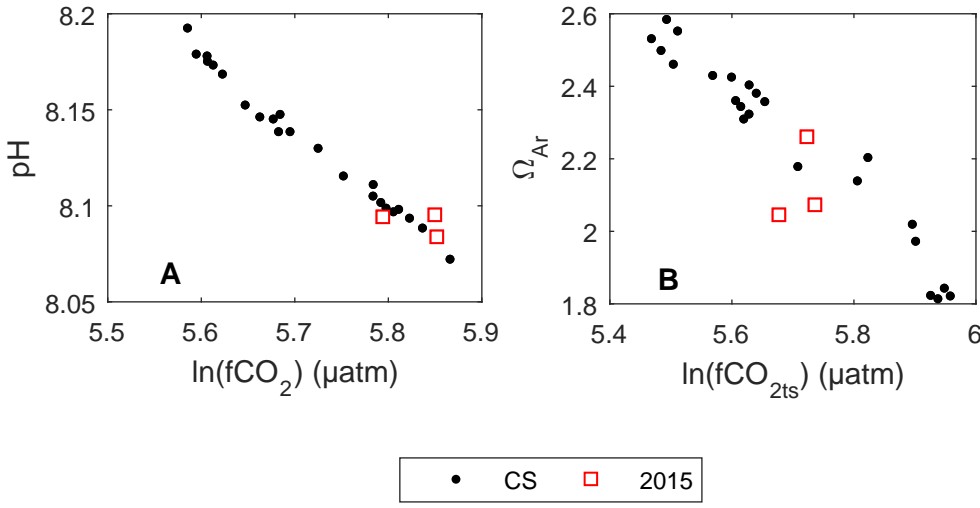