# Peer review of "Aragonite saturation states and pH in western Norway fjords: 1 seasonal cycles and controlling factors, 2005-2009 2 3 A. M. Omar1,2, I. Skjelvan1, S.R. Erga3, A. Olsen2 4 5 1: Uni Research Climate, Bjerknes Centre for Climate Research, Bergen, N"

_Ocean Science, 2016_

## Referee Comment (RC1) · M. Ribas-Ribas (Referee) · 11 Apr 2016

**Comment on "Aragonite saturation states and pH in western Norway fjords: seasonal cycles and controlling factors, 2005-2009" by A. M. Omar et al.**

**General comments:**

The present paper under review for Ocean Science describes seasonal cycles and controlling factors on Aragonite saturation states and pH and proposes a nice way to monitore Ocean Acidification from continuous underway $p\mathrm{CO}_2$ measurements. I appreciate the effort to maintain the VOS line for so long and the extra sampling cruise. Together, makes a valuable dataset! It is also really easy to read as it flows. I will recommend the publication after moderate corrections/clarifications.

However, I have two major concerns about the current version of the manuscript:
1. Statistical analysis need to be better explained and improved to have more robust conclusion. For example, fig. 2 B (8-249, 9-264) explain that they are UW and RF are really similar but I would like to know the slope and if this slope is statistically different than 1 (same apply for B and D). I would add other stats comments and the list of general comments.
2. I would like to know more about interannual variability. This is also somehow what I expect after reading introduction (3-91). But still we only have some sentences at the end of section 3.3. Related to this, I don't understand why you choose VOS data since 2009 but then you have CS since 2010, a cruise in 2015 and a sensor in 2012. I understand it's not easy to treat data from VOS as quick as they are generated but at least some words of cautions will be nice.

**Specific comments:**

My comments will be with format page-line (so 5-3 means page 5, line 3); page-line:line (so 7-7:9 means page 7 from line 7 until line 9).

1-17 What coherent means in this contest?

1-21 (and elsewhere in the ms) Please choose as many decimal place you want to report according to your method accuracy and error propagation and stick to them. Here you start with 2 and then past to 1 but further in the ms, suddenly there is 4-5 digits (9-272, 10-284).

2-36 What is Peters reference referring to? It looks strange.

2-51-54 Long paragraph

2-61 "Only few studies" I expect at the end of the sentence some citation

3-75:80 I think that should flow better with a bit of rearranging to avoid repetition together with 1-61:2-68

I really like this part where it is explain the importance of the study, to "fill the gap". It is indeed extremely important to fill these coastal gaps. I think it will also be nice to mention that this study set the baseline for future studies to evaluate OA and how this change could affect.

5-130 "coastal open ocean" is somehow misleading, although I understand might be good to rephrase.

5-143:146 Economic important could be also tourist and other ecosystem functions?

5-149 "of" is double $CO_2$ is without subscript (also in equation 3 and 4 and 12-360 and some references)

5-151 webpage without hyperlink, like for example 6-166 (some happens in 7-197)

6-166 knowing the authors, I suppose there are plan to submit the remaining data to SOCAT, right?

6-176 (and elsewhere in the ms, for example table 1 or 7-195), unify dates format

7-197 It will be nice to state here a few details from sensor, like accuracy or/and precision, as it is important to propagate the other uncertainties in the calculations.

7-217 Here an important point, TA-SSS algorithm comes only from summer values, right? Please clarify and add some sentence of caveat about it.

8-222 Which other ancillary parameters? Where they come from?

8-244 You haven't describe RMS before. Salinity is undimension. It will be nice to have some idea here of what this 0.81 could means in terms of pH or OmegaAr, with initial error propagation

10-281 Fig. 2 should read Fig. 3. Also normally better to start the sentence in more direct way. I would like to know more details how is this data collapsed: what we see is means? What about std? This comment is together with this interannual variation maybe.

10-284 text said maximum is 8.2 but figure goes until 8.35

I like this comparison with C-CAN ☺

10-311 Could you add a sentence in methodology or somewhere explaining how DIC is normalize, which salinity has been used, which reference? Also note OmegaAr (sst) has something between t and ). Also extra space before brakets ini 11-313

11-314 State the table or fig that is showing that (I think Fig. 4)

11-315 Maybe nice to state the basics about Lauvset methods. Also note extra brakets before Lauvset and missing before the year.

11-316 Left panels maybe better for state the letters (a, c…)

11-319 Fig. 4 c, d, e (mismatch of figure number, spaces missing after comma and letters in the figure are capital and in the test are not, please unify). Also amend for 11-321,11-333, 12-369 and 11-322.

11-324:325 This sentence need more explanation. Although I agree, somehow contradict the idea of if we measure $p$CO$_2$ and temperature we can estimate OA parameters.

11-329 Fig. 4 (B, D)

11-332 Refer to the figure (Fig. 4G and salinity data not shown)

11-334 I think that should read Fig. 4F

After reading this twice, an idea can be to do some "Multiple linear regression" to support your analysis in a statistical way.

11-337 impact. (extra space before point) Also in 11-342 before sst).

12-359 equations 5 and 6 or equation 3 and 4 (somehow not easy to understand)

12-366 another stats comment, how significantly is this significantly?

12-371 there are two commas before R2

13-372 there is extra point after (6)

13-373 or somewhere else, just though that another importance to use VOS $p$CO$_2$ is that they are more frequent that normal oceanography cruises in winter, especially in these regions.

A note of caution about the slopes, we need some +- because if we took the slope of red square will be really different. But still I like the comparison with RCP scenarios.

13-387 cannot

13-401 extra : at the end of captions

14-420:423 I miss some words of what this low omegaAr 1.3 could mean for calcifying organism. Some studies point that there's no need to be below 1 to have some affect in organism. Some "bio"-words could expand the readers-users of these nice results.

14-424:425 Maybe I miss something but this sentence is not really new or surprising, right?

Table 1 What's the different between 1 and 5 m. Are they both underway water?

22-607 Should be table 2

Figure 1 may benefit from a more global map, for no European readers. Also if possible add the meaning of the arrows (NCC) in the figure, so we don't have to read legend.

Fig. 2A, is colour scale and x-axis the same SST

Fig. 3 is missing A-D and parameter and unit on the colour scale. Also unify decimal units

Fig. 4 You explain good that SST and TA are not independent. So question is, could you still use this method with dependent variables? Also careful because labels are cutting x-axis

---

## Referee Comment (RC2) · E.M. Jones (Referee) · 27 Apr 2016

This study presents seasonal cycling and controlling factors of the ocean acidification parameters, aragonite saturation and pH, in fjords of western Norway. The data and discussions are an important contribution to CO2 research in a relatively undersampled region and the methods used to determine the whole carbonate system from underway pCO2 data form a novel approach and are complimentary to future studies in this field. The choice of the journal fits very well and I recommend publication of this article. In the version reviewed here there are a number of ameliorations that can be made in terms of clarity, coherence and rigour before proceeding with publication. The authors may find comments, questions and linguistic corrections analytically below.

[Figure]

General comments:

1. A very minor comment regarding language; as it is a European journal I would suggest replacing the use of "fall" with "autumn".

2. Make sure it is clear how TA is derived from SSS(sst) and SST, i.e, how strong is a relationship between SSS and SST to make such a derivation and secondly what steps are carried out to determine TA?

3. Include a clear statement of the precision and accuracy of the measured parameters. This would fit well before the detailed description of the propagation of errors and further place them in context.

4. The influence of sea ice and glacial ice dynamics has not been explicitly considered. Is this area free from ice influences and if so perhaps state this or does sea ice formation in winter and melt during spring-summer contribute to the correlation in SST and SSS?

5. For salinity normalised DIC, the terms DICS and nDIC are both used. Perhaps try and use one term for consistency and clarity throughout the paper and include a statement of how the normalisation was carried out, i.e., what salinity reference is used, if the standard correction was made or non-zero endmembers were included, as described in Friis et al. (2003).

6. Figure 2: in relation to the various regression analyses made, a statement on the interpretation of the regression and the significance of the relationships is required. This would make the comparisons of the parameters more relevant and strengthen the choice of those used in other calculations.

7. Check Table numbering; Table 2 is absent.

Specific comments:

Abstract

Line 9 write "carbon dioxide (CO2)" in full (first time use in main text). Line 10 insert "the" between "lowers" and "aragonite saturation state". Line 19 change to "carbon-rich". Line 24-25 replace "brings up" with "enables" and add "reach" between "to" and "the". Line 32 define "SSS" for first time use.

1. Introduction

Line 38 replace "incur" with "cause". Line 39 replace "ocean" with "oceanic". Line 43 insert "," after the reference. Line 51 insert "," after "regions". Line 63 insert "a" between "is" and "prerequisite".

Line 64 remove the "s" on "latitudes". Line 65 insert "CO2" between "be" and "sources". Line 76 delete "important" to refrain from using the same word twice in one sentence. Line 81 delete "the" between "in" and "surface". Line 89 insert "the" between "present" and "mean".

1.1 The Study Area

Line 93 instead of capital letters use lower case "n" and "s" for "north" and "south", respectively. Line 95 replace "at" with "along".

Line 103 delete "the" at the end of the line. Line 106 replace "it" with "Korsfjord" to reduce the use of generic "it" and be specific to the subject under description. Line 108 remove full-stop and replace with and to join sentences for better flow. Line 109 correct "witch" to "which". Line 118 replace "mix" with "mixes". Line 119 insert second

bracket ")" after "(NCC". Line 121 change "wind" to "winds" and insert "the water in" after "circulation of". Line 122 replace "that follows" with "following". Line 124 suggestion to add "forcing" after "wind". Line 125 replace "," with "and inputs of". Line 125-126 between "seasonal time scales," and "during spring-summer" replace text with "salinity drives stratification". Line 126 add "the water column is" between "and" and "more".

Line 129 specify what "the temperature" refers to, i.e., a certain depth or depth range in the water column or certain location in the fjord. Line 131 delete "the" start line with "Water". Line 132 following "oxygen" insert "to the area". Line 132-133 delete "the fjords enhance their" and insert "is enhanced in the fojrd" between "production" and "which". Line 133 replace "enables them to host" with "supports". Line 137 replace "decisive" with "dominant controls of" and delete "for the". Line 140 write "nutrient-rich" and add "on from" after "follows". Line 142 insert "upper" before "water column", i.e., assuming that the sub-surface/ deeper waters remain nutrient rich? Line 143 insert 2 "," either side of "with its adjacent waters". Line 149 replace the extra "of" with "the" and make the 2 on "CO2" a subscript. Line 152 replace "It" with "The ship". Line 153 replace "." after "(Fig. 1)" with "," and delete "It".

Line 6

Line 161 replace "in" with "during". Line 166 add SOCAT reference: Bakker, D. C. E., Pfeil, B., Smith, K., Hankin, S., Olsen, A., Alin, S. R., Cosca, C., Harasawa, S., Kozyr, A., Nojiri, Y., O'Brien, K. M., Schuster, U., Telszewski, M., Tilbrook, B., Wada, C., Akl, J., Barbero, L., Bates, N. R., Boutin, J., Bozec, Y., Cai, W.-J., Castle, R. D., Chavez, F. P., Chen, L., Chierici, M., Currie, K., De Baar, H. J. W., Evans, W., Feely, R. A., Fransson, A., Gao, Z., Hales, B., Hardman-Mountford, N. J., Hoppema, M., Huang, W.-J., Hunt, C. W., Huss, B., Ichikawa, T., Johannessen, T., Jones, E. M., Jones, S., Jutterstrom, S., Kitidis, V., Körtzinger, A., Landschützer, P., Lauvset, S. K., Lefèvre, N., Manke, A. B., Mathis, J. T., Merlivat, L., Metzl, N., Murata, A., Newberger, T., Omar,

[Figure]

A. M., Ono, T., Park, G.-H., Paterson, K., Pierrot, D., Ríos, A. F., Sabine, C. L., Saito, S., Salisbury, J., Sarma, V. V. S. S., Schlitzer, R., Sieger, R., Skjelvan, I., Steinhoff, T., Sullivan, K. F., Sun, H., Sutton, A. J., Suzuki, T., Sweeney, C., Takahashi, T., Tjiputra, J., Tsurushima, N., Van Heuven, S. M. A. C., Vandemark, D., Vlahos, P., Wallace, D. W. R., Wanninkhof, R., Watson, A. J. (2014) An update to the Surface Ocean CO2 Atlas (SOCAT version 2). Earth System Science Data 6: 69-90. doi:10.5194/essd-6-69-2014. Line 169 replace "in the period" with "during". Line 171 write "CarboSchools (CS)" in full (first time use in main text). Line 186 write "Conductivity Temperature Depth (CTD)" in full (first time use in main text) and maybe specify what "data" was acquired and/ or used in this study.

Line 192 instead of capital letter use lower case "s" for "station". Line 200 clarify the estimated pH values – estimated from what source? CO2SYS?

2.4 Methods

Line 206-207 insert "an" before "empirical" and define the parameters used to derive the empirical relationship. Line 217 replace "Second" with "Secondly". Line 217 clarify how TA is derived from SSS(sst), see general comments.

3. Results and discussion

3.1 Correlations and validations

Line 236 insert "the" before "data". Line 237-239 Is sea ice present in the region? see general comments. Line 239 clarify where the "runoff" originates. Line 240 insert "degree of" between "high" and "scatter". Line 244 reverse order of "indeed is" to "is indeed". Line 247 delete "a" after "As" and re-write sentence following "verification"

as "that the RF SST dataset is spatially representative,". Line 249 replace "in" with "across". Line 256 replace "statics" with "statistics". Line 257 replace "functions" with "acts" and insert "an" between "as" and "indicator". Line 259 add a space to separate "with" and "values". Line 260 add commas to read "data ,i.e., pH(sss)". Line 265 insert "closely" between "$\Omega$Ar(sst)" and "reproduce" and delete "very well". Line 269 replace "show" with "shown". Line 274 insert "through the calculations" between "propagated" and "to".

**3.2 Spatiotemporal variations**

Line 281 replace "collapsed" with "condensed". Line 284 delete "the" following "after". Line 286 replace "outweighs" with "counteracts". Line 287 replace "begins" with "begin" and also specify what processes are being referred to. Line 288 insert "the" after September replace "SSTs" with "SST". Line 289 re-phrase sentence between "mixing," and "as mentioned" to read "which enables deep, carbon-rich coastal water to penetrate the surface layer,". Line 290 replace "as" between "and" and "reflected" with "is" and delete "the" before "increasing". Perhaps specify which DIC values are increasing by the autumnal mixing, i.e., sea surface, surface layer, upper ocean... Line 293 replace "drives up the " with "enriches". Line 294 insert "the" before "concentration", delete "the" between "of" and "carbonate" and add an s to read "ions". Line 296 re-phrase to read "due to inputs of run-off" and replace "reinforces" with "enhances". Line 297 replace "over" with "exhausted". Line 298 replace "The" with "However," and delete ", on the other hand,". Line 300 replace "mismatch" with "decoupling of". Line 307 correct "embody" to "embodied".

**3.3 Controls of seasonal variability and trends**

Line 310 replace "arranged the data into" with "computed". Page 11 Line 315 re-write

the reference to "Lauvset et al. (2015)". Line 316-317 replace "are shown on Fig. 4 (left panels) where it can be seen" with "show" – removes the double Fig. reference in the sentence. Line 326 replace "letting one of the drivers" with "varying them independently". Line 327 delete "to vary" and replace "drivers" with "controls" – removing double word use in the same sentence. Line 329 replace "for" with "of" and correct "driver" to "drivers". Line 331 correct "induce" to "induces". Line 332 insert "other" between "the" and "hand". Line 333 correct "induce" to "induces". Line 334-335 re-write to "We therefore conclude that variations in DIC, followed by TA, are the most important drivers for changes in $\Omega Ar(sst)$." Line 337 replace "have" with "has" and delete "In terms of processes" so that the next line starts with "This means". Line 338 correct to "carbon-rich". Line 339 insert "inputs" after "runoff". Line 343 replace "changes" to "change". The first use of nDIC – how is defined and how are the normalised values determined, perhaps add reference, e.g., Friis et al. (2003).

3.4 Inference of OA parameters from VOS underway data

Line 353 the use of DICS is not specifically defined in the text and is an additional term to nDIC – do they both refer to salinity normalised DIC determined in the same way? Please clarify to be sure. Line 355 remove the space after (2010). Line 359 reference to equations 4 and 5 might not be correct, i.e., should it be equations 3 and 4. Please check and amend if necessary. Line 362 remove ", while the CO2 system is fully determined only occasionally," as I don't think it adds anything it would reduce the length of the sentence. Line 368 delete the spare "and" after "Fig. 5" and add "to" between "conform" and "tight". Line 371 replace extra "," with ";" after "10.354" in equation 5.

Line 372 check for extra spaces and uses of "," and/ or ";" in separation of terms and consistency with equation 5. Line 377 delete "and" before "thus" and replace "minimise" with "minimizing". Line 380 replace "with" by "to". Line 391 change to "carbon-rich". Line 392 rearrange the line to read "surface water also reflects the properties of the deeper water". Line 394 replace "can" with "could" and insert "to occur" after "expected". Line 399 replace "development" with "CO2 concentrations" and replace "follows" with "follow". Line 400 add "driving oceanic CO2 uptake" after "atmosphere".

4. Summary and concluding remarks

Line 402 insert "worth" after "four years" and replace "less" with "sporadic". Line 403 delete "frequent". Line 413-414 use the same number of decimal places for quoted numbers. Line 415 insert "the" before "phytoplankton" and replace ", which" with "that" and delete "levels". Line 417 replace "brings up" with "allows"; correct to "carbon-rich"; insert "mix into" between "to" and "the". Line 418-419 replace "and reinforces the decrease in pH" with "reduces" and delete ", which continuous throughout fall". Line 430-431 finish sentence after "carbonate system".

Line 435 replace "development" with "concentrations" and remove "f" from "fCO2", i.e., referring to concentrations of CO2 rather than the concentrations of the fugacity of CO2 that wouldn't be quite correct. Line 436 finish the sentence after "atmosphere".

5. Acknowledgements

Line 438 replace "supports" with "support" and replace "by" with "from". Line 440 insert "the" after "and help of".

Tables

Table 1
Page 20 Line 598 replace "about" with "of". Line 599 insert "which" after "for".

Page 21 Line 603 replace "(continue)" with "(continued)".

Table 3

Page 22 nDIC – "DIC normalised to the mean salinity"; please include somewhere in the main body of the text what value was determined for the mean salinity and how.

Figure texts

Page 23 Line 620 replace "which" with "where" and replace "tick" with "thick". Line 635 check "ln(nfCO2ts)" – is the "n" correct and currently it doesn't match with the X-axis labelling in Figure 5 (B).

Figures

Figure 2 (A) Y-axis can be expanded to open up the visual regression trend, i.e., range in SSS of 25-35. Confusion of the colour bar underneath is too close to the X-axis and on first inspection looks like it is referring to SST values. Add "month" as the unit to clarify the colour bar and move it closer to panel (A) if possible. Figure (3) 4 time series panels are presented but only 2 parameters mentioned in the Figure text and all colour bars are unlabelled. Add parameter labels and units (where relevant) to each of the coloured panels.

---

## Author Comment (AC1) · 28 Apr 2016

We thank the reviewers (Dr Mariana Ribas-Ribas and Dr. E.M. Jones) for encouraging and constructive comments as well as suggested improvements. In their "general comments" the two referees asked for improved presentation of the statics and interpretation of the regressions and more elaboration on (i) the interannual variations and (ii) the salinity normalization of the dissolved inorganic carbon (DIC). We agree with the reviewers in all the above points. We will therefore address all the issues raised by the referees (general as well as specific) in the revised version of the manuscript which will be submitted in due time. A detailed response to the referee comments will accompany the revised version.

Best Regards, on behalf of all co-Authors Abdirahman M. Omar

---

## Author Response (AR1)

**Combined authors response and changes-tracked revised manuscript of OS\_2016\_09.**

In the following we provide a detailed point-by-point response to all referee comments and specify all changes in the revised manuscript. The response to the Referees includes comments from Referees (in black) and author's response and author's changes in manuscript (in blue). In addition, we provide a marked-up manuscript version showing the changes.

**Reviewer 1 (Dr. M. Ribas-Ribas).**

The present paper under review for Ocean Science describes seasonal cycles and controlling factors on Aragonite saturation states and pH and proposes a nice way to monitore Ocean Acidification from continuous underway pCO2 measurements. I appreciate the effort to maintain the VOS line for so long and the extra sampling cruise. Together, makes a valuable dataset! It is also really easy to read as it flows. I will recommend the publication after moderate corrections/clarifications.

Response: We thank the reviewer for encouraging and constructive comments as well as for suggested corrections and improvements.

However, I have two major concerns about the current version of the manuscript: 1. Statistical analysis need to be better explained and improved to have more robust conclusion. For example, fig. 2 B (8-249, 9-264) explain that they are UW and RF are really similar but I would like to know the slope and if this slope is statistically different than 1 (same apply for B and D). I would add other stats comments and the list of general comments.

Response: We agree with the reviewer on this point.

In the revised manuscript, the statistics of all comparisons are presented. An additional table (Table 3) summarizing the statistics of the comparisons on Fig. 2 is included. Further, in order to introduce how these statistics is used the following paragraph has been added to the beginning of section 3.1 "In this section we present the regression equations identified in this study in addition to validating the various estimation procedures used by comparing the estimated values with those measured/computed. The results of these comparisons are summarized in Table 3. For each comparison, the coefficient of determination ( $R^2$ ) and the significance level (p-value) are used as metrics for the goodness of the correlation while the associated root-mean-square error (rmse) is benchmarked against the maximum target uncertainties developed by the Global Ocean Acidification Network (GOA-ON) and the California Current Acidification Network (C-CAN) of  $\pm 0.2$  for  $\Omega_{Ar}$  (McLaughlin et al., 2015), which corresponds to maximum uncertainties of  $\pm 0.02$ ,  $\pm 1.25$  or  $\pm 1.8$  in pH, SST, or SSS, respectively."

2. I would like to know more about interannual variability. This is also somehow what I expect after reading introduction (3-91). But still we only have some sentences at the end of section 3.3. Related to this, I don't understand why you choose VOS data since 2009 but then you have CS since 2010, a cruise in 2015 and a sensor in 2012. I understand it's not easy to treat data from VOS as quick as they are generated but at least some words of cautions will be nice.

Response: we agree with the reviewer to write more about the interannual variations (IAV), although the focus of the study is on the seasonal variations.

In the revised manuscript, IAV have been computed as the standard deviations of monthly means. The results are summarized in additional table (Table 4) and results commented in the end of section 3.3.

Regarding the temporal coverage of the datasets, we used all underway data that has been available from the VOS line which was in operation in the period September 2005 to September 2009. Cruise data with dates beyond the operating period of the VOS line has been used in order to maximize the data (in terms of number of data points as well as in spatial and temporal coverage) available from the study area to identify the regional SSS/SST – Alkalinity relationship (Eq. 2 in the manuscript). In order to clarify the above points, the first paragraph of section 2.1 in the revised ms, includes the sentence "*The VOS line was in operation in the period September 2005 to September 2009*."

My comments will be with format page-line (so 5-3 means page 5, line 3); page-line:line (so 7-7:9 means page 7 from line 7 until line 9).

1-17 What coherent means in this contest?

Response: with "Spatially Coherent" we mean that seasonal changes with similar magnitudes occur everywhere at the same time.

1-21 (and elsewhere in the ms) Please choose as many decimal place you want to report according to your method accuracy and error propagation and stick to them. Here you start with 2 and then past to 1 but further in the ms, suddenly there is 4-5 digits (9-272, 10-284). Response: in the revised ms, pH values are presented with three decimal points while two decimals are used for the saturation state.

2-36 What is Peters reference referring to? It looks strange.

Response: this has been replaced with more relevant reference (Le Quéré et al., 2015) for updated assessment of global emissions.

**2-51-54 Long paragraph**

2-61 "Only few studies" I expect at the end of the sentence some citation

Response: in the revised ms (page 3), this sentence has been extended to "*However*, only a few studies on the carbon cycle of Norwegian fjords exist in the literature, and these are from the high Arctic at Svalbard (Fransson et al., 2014; Omar et al., 2005)."

3-75:80 I think that should flow better with a bit of rearranging to avoid repetition together with 1-61:2-68

I really like this part where it is explain the importance of the study, to "fill the gap". It is indeed extremely important to fill these coastal gaps. I think it will also be nice to mention that this study set the baseline for future studies to evaluate OA and how this change could affect.

Response: we thank the reviewer for a good suggestion.

In the revised ms (page 3), we have rearranged the first sentence as suggested. We have also extended the paragraph with "Observations of the carbon cycle dynamics in the fjord system will not only further our understanding and ability for prediction, but they will also serve as benchmarks against which future changes are compared."

5-130 "coastal open ocean" is somehow misleading, although I understand might be good to rephrase.

Response: in the revised ms, the phrase "coastal open ocean" has been replaced with "adjacent coastal North Sea".

5-143:146 Economic important could be also tourist and other ecosystem functions? Response: we tried to stick to the natural ecosystems only. However, we agree that we needed to elaborate the economic significance. Thus, in the revised ms (end of section 1.1), we extended the paragraph with "*The fjord system also contributes to the important aquaculture production that, with its annual fish production of* >700 *tonnes, ranks Norway within the tenth place worldwide. About one fifth of this is produced in the Hordaland County where the fjord system studied here is situated (http://www.diercke.com).*"

5-149 "of" is double CO2 is without subscript (also in equation 3 and 4 and 12-360 and some references)

Response: corrected.

5-151 webpage without hyperlink, like for example 6-166 (some happens in 7-197) Response: corrected.

6-166 knowing the authors, I suppose there are plan to submit the remaining data to SOCAT, right?

Response: the reviewer is right. Actually, the data is submitted as mentioned in revised ms (last sentence of section 2.1).

6-176 (and elsewhere in the ms, for example table 1 or 7-195), unify dates format Response: done.

7-197 It will be nice to state here a few details from sensor, like accuracy or/and precision, as it is important to propagate the other uncertainties in the calculations. Response: done.

7-217 Here an important point, TA-SSS algorithm comes only from summer values, right? Please clarify and add some sentence of caveat about it.

Response: we agree with the reviewer.

In the revised ms, we have discussed this issue in connection with the comparison between wintertime pH estimates and in situ pH measurements from January 2012. In the end of section 3.1 we conclude "It must be noted that the above total error was derived from all available observational data including the in situ sensor data (shown in Fig. 2c and in described section 2.3), which are the only wintertime measurements used in this study. This is important because the lack of wintertime data in the CS dataset which was used for the identification of AT-SSS/SST relationship (Eq. 2) means that wintertime AT(sst) might be overestimated so that corresponding pH(sst) values would be overestimated. In fact, during the aforementioned comparison between pH(sst) and measured pH we noted that for this particular dataset pH(sst) overestimated the measurements. However, the estimates were consistent with the observations to within the total error of  $\pm 0.01$  pH units. Thus, by utilizing the above total errors, we also accounted for the effect of this possible caveat of Eq. 2 arising from the lack of wintertime TA measurements."

8-222 Which other ancillary parameters? Where they come from?

Response: in the revised ms (page 8), this sentence has been modified to "*The UW fCO2* together with TA (sst), UW SST, and SSS(sst) were then used to characterize the full seawater CO2 chemistry using CO2SYS (Lewis and Wallace, 1998; van Heuven et al., 2011), with K1 and K2 constants from Lueker et al. (2000)."

8-244 You haven't describe RMS before. Salinity is undimension. It will be nice to have some idea here of what this 0.81 could means in terms of pH or OmegaAr, with initial error propagation

Response: in the revised ms, statistics of equations and comparisons clarified, and rmse is defined. Additionally, the errors in pH, SST, and SSS that can produce the maximum target uncertainty of  $\pm 0.2$  for  $\Omega_{Ar}$  developed by the C-CAN and GOA-ON have been estimated (see 1st paragraph of section 3.1).

10-281 Fig. 2 should read Fig. 3. Also normally better to start the sentence in more direct way. I would like to know more details how is this data collapsed: what we see is means? What about std? This comment is together with this interannual variation maybe.

Response: what we show on Fig. 3 is gridded data based on weighted averages. In the revised ms (section 3.2, and tables), a table showing the interannual variations has been added (Table 3). Additionally, the first paragraph of section 3.2 is modified to "In order to present the mean distributions across the different fjords throughout the annual cycle, we collapsed the data into one virtual year by projecting it onto non-equidistant rectangular grids using the "weighted-average gridding" method of the Ocean Data View software (Schlitzer, 2015). As evident from Fig. 3, there is a clear seasonality in both pH(sst) and  $\Omega ar$  (sst)."

10-284 text said maximum is 8.2 but figure goes until 8.35

Response: winter pH values are typically about 8.25, but extreme pH values of >8.3 occurred during the winter of 2008.

In the revised ms (section 3.2, and tables), this is clarified by stating that pH "...varies between minimum values (8.05) around New Year to typical maximum values of around 8.25, which occur during the late winter and/or spring (March-April)." Additionally, a table showing the interannual variations has been added (Table 3).

I like this comparison with C-CAN  $\blacksquare$

10-311 Could you add a sentence in methodology or somewhere explaining how DIC is normalize, which salinity has been used, which reference? Also note OmegaAr (sst) has something between t and ). Also extra space before brakets ini 11-313 Response: corrected.

11-314 State the table or fig that is showing that (I think Fig. 4) Response: done as suggested by the reviewer.

11-315 Maybe nice to state the basics about Lauvset methods. Also note extra brakets before Lauvset and missing before the year.

Response: done as suggested by the reviewer.

In the revised ms (page 12), we added the sentence "*This method estimates the monthly pH changes expected from corresponding changes observed in SST, SSS, DIC, and TA as well as their sum.*"

11-316 Left panels maybe better for state the letters (a, c...) Response: done as suggested by the reviewer. 11-319 Fig. 4 c, d, e (mismatch of figure number, spaces missing after comma and letters in the figure are capital and in the test are not, please unify). Also amend for 11-321,11-333, 12-369 and 11-322.

Response: corrected as suggested by the reviewer.

11-324:325 This sentence need more explanation. Although I agree, somehow contradict the idea of if we measure  $pCO_2$  and temperature we can estimate OA parameters.

Response: The sentence is not about our ability to estimate OA parameters. It is about our ability to analyze the controls of OA parameters for which measured SSS and AT are best. If estimated AT(sst) is used, on the other hand, the magnitude of the alkalinity control will be more uncertain while its form will resemble that of SST.

In the revised ms (page 13), the sentence is extended to "*This emphasizes the need for* measured SSS and TA values when the objective is to analyze the controls of pH and  $\Omega_{Ar}(sst)$  variations."

11-334 I think that should read Fig. 4F

After reading this twice, an idea can be to do some "Multiple linear regression" to support your analysis in a statistical way.

Response: here we disagree with the reviewer. First, "Fig. 4" is correct i.e. that is where the seasonal amplitude of  $\Omega_{Ar}(sst)$  is shown. Secondly, we have already verified that multiple linear regressions are not a better alternative for our analyses. Such regression would only give us indication on how well the  $\Omega_{Ar}(sss)$  values can be modelled from a given set of parameters, but the values of the regression coefficients would NOT necessarily reflect sensitivity of  $\Omega_{Ar}(sss)$  to the associated parameters.

11-337 impact. (extra space before point) Also in 11-342 before sst).

Response: corrected as suggested by the reviewer.

12-359 equations 5 and 6 or equation 3 and 4 (somehow not easy to understand) Response: we agree with the reviewer. The ambiguity arised as result of a typo. In the revised ms (page 14), the sentence is corrected to "*This, together with the fact that equations 3 and 4 can...*"

12-366 another stats comment, how significantly is this significantly?

12-371 there are two commas before R2

13-372 there is extra point after (6)

Response: Thank you! Both the above typos have been corrected in the revised text.

13-373 or somewhere else, just though that another importance to use VOS pCO2 is that they are more frequent that normal oceanography cruises in winter, especially in these regions. Response: Good idea! This is implemented in the revised manuscript (page 15).

A note of caution about the slopes, we need some +- because if we took the slope of red square will be really different. But still I like the comparison with RCP scenarios. Response: we agree with the reviewer.

In the revised ms (end of section 3.4), an uncertainty of  $\pm 0.2$  in the  $\Omega_{Ar}$  estimates is included and the text has been changed accordingly.

**13-387 cannot**

Response: we are not sure what this comment means, but the word "cannot" was used intentionally in the original manuscript.

13-401 extra : at the end of captions Response: corrected.

14-420:423 I miss some words of what this low omegaAr 1.3 could mean for calcifying organism. Some studies point that there's no need to be below 1 to have some affect in organism. Some "bio"-words could expand the readers-users of these nice results. Response: ....

14-424:425 Maybe I miss something but this sentence is not really new or surprising, right? Response: in the revised text (page 17), this sentence has been modified to "Strong correlations of pH and  $\Omega_{Ar}$  with  $fCO_2$  and  $fCO_{2@meanSST}$  ( $fCO_2$  adjusted to the mean temperature), respectively provide an approach to interpolate pH and  $\Omega_{Ar}$  over large areas in the fjords of western Norway where underway measurements of  $fCO_2$ , SST, and SSS are available."

Table 1 What's the different between 1 and 5 m. Are they both underway water? Response: The RF dataset are averages over measurements five depths 1-5 m. This is clarified in the revised manuscript (page 7, and Table 2 and its caption) 22-607 Should be table 2 Response: corrected.

Figure 1 may benefit from a more global map, for no European readers. Also if possible add the meaning of the arrows (NCC) in the figure, so we don't have to read legend. Response: a more global map is used in the revised manuscript.

Fig. 2A, is colour scale and x-axis the same SST Response: No, the color scale shows months. This is clarified in the revised manuscript (Fig. 2A).

Fig. 3 is missing A-D and parameter and unit on the colour scale. Also unify decimal units Response: The missing legends "A-D" has been added to Fig. 3 of the revised manuscript.

Fig. 4 You explain good that SST and TA are not independent. So question is, could you still use this method with dependent variables? Also careful because labels are cutting x-axis. Response: Yes, we can use this procedure even if TA is based on SST data. This is because the decomposition procedure computes the effect of changes in each driver (i.e. changes in SST, SSS, TA, and DIC) separately. Moreover, the seasonal changes in TA and SST are opposite i.e. TA decreases when SST increases and vice versa, and the magnitude of the former is about ten times bigger. Thus, even though the similarity of Fig. C and E (in from) is as expected, the magnitudes of the resulting effects from the two drivers can be different (these are currently nearly equal by coincidence) if, for instance, we use a TA Revelle Factor slightly different form the global value of -10.

**Dr. Reviewer 2 (E.M. Jones)**

Comment:

"This study presents seasonal cycling and controlling factors of the ocean acidification parameters, aragonite saturation and pH, in fjords of western Norway. The data and discussions are an important contribution to CO2 research in a relatively undersampled region and the methods used to determine the whole carbonate system from underway pCO2 data form a novel approach and are complimentary to future studies in this field. The choice of the journal fits very well and I recommend publication of this article. In the version reviewed here there are a number of ameliorations that can be made in terms of clarity, coherence and rigour before proceeding with publication. The authors may find comments, questions and linguistic corrections analytically below."

Response: We thank the reviewer for these encouraging and constructive comments as well as for the suggested corrections and clarifications.

C: 1. A very minor comment regarding language; as it is a European journal I would suggest replacing the use of "fall" with "autumn". Response: done as requested by the reviewer.

2. Make sure it is clear how TA is derived from SSS(sst) and SST, i.e, how strong is a relationship between SSS and SST to make such a derivation and secondly what steps are carried out to determine TA?

Response: this has been clarified as suggested by the reviewer.

In the revised manuscript (page 10), it is stated "To estimate a corresponding TA value for each UW fCO2 observation obtained from MS Trans Carrier, we used salinity values estimated from UW SST data by using Eq.1. The results (denoted as SSS(sst)) were then inputted into Eq.2 to obtain TA(sst) (Table 2). The fact that TA(sst) are based on SSS(sst) rather than measured SSS values introduced an additional error in the estimated pH(sst) and  $\Omega Ar(sst)$ . In order to assess this error...."

3. Include a clear statement of the precision and accuracy of the measured parameters. This would fit well before the detailed description of the propagation of errors and further place them in context.

Response: in the revised ms, the measurements accuracy of DIC, TA, and pH is stated (section 2.2 and 2.3).

4. The influence of sea ice and glacial ice dynamics has not been explicitly considered. Is this area free from ice influences and if so perhaps state this or does sea ice formation in winter and melt during spring-summer contribute to the correlation in SST and SSS?

Response: the study area is free from ice influence.

In the revised ms (section 1.1), this is clarified by stating "...snow melt and run-off are the main local sources for freshwater since the fjord system is generally ice free."

5. For salinity normalised DIC, the terms DICS and nDIC are both used. Perhaps try and use one term for consistency and clarity throughout the paper and include a statement of how the normalisation was carried out, i.e., what salinity reference is used, if the standard correction was made or non-zero endmembers were included, as described in Friis et al. (2003).

Response: "DICS" was found in Eqs. 3 and 4 in the original the manuscript. However, this is referring to the product "DIC" times the parameter "S". The latter has been defined in Egleston et al. (2010) as referenced in the text.

In the revised manuscript, all abbreviations used DIC, nDIC are properly defined. Additionally, an overview of the symbols used for quantities estimated and/or derived from the measurement-based variables is given on Table 2. 6. Figure 2: in relation to the various regression analyses made, a statement on the interpretation of the regression and the significance of the relationships is required. This would make the comparisons of the parameters more relevant and strengthen the choice of those used in other calculations.

Response: we thank the reviewer for the suggested improvement.

In the revised manuscript, the statistics of all equations and comparisons are presented, and an additional table (Table 3) summarizing the statistics of the comparisons on Fig. 2 is included.

7. Check Table numbering; Table 2 is absent.

Response: corrected.

Specific comments:

Abstract

Line 9 write "carbon dioxide (CO2)" in full (first time use in main text). Line 10 insert "the" between "lowers" and "aragonite saturation state". Line 19 change to "carbonrich". Line 24-25 replace "brings up" with "enables" and add "reach" between "to" and "the". Line 32 define "SSS" for first time use.

1. Introduction

Line 38 replace "incur" with "cause". Line 39 replace "ocean" with "oceanic". Line 43 insert "," after the reference. Line 51 insert "," after "regions". Line 63 insert "a" between "is" and "prerequisite".

Line 64 remove the "s" on "latitudes". Line 65 insert "CO2" between "be" and "sources". Line 76 delete "important" to refrain from using the same word twice in one sentence. Line 81 delete "the" between "in" and "surface". Line 89 insert "the" between "present" and "mean".

1.1 The Study Area

Line 93 instead of capital letters use lower case "n" and "s" for "north" and "south", respectively. Line 95 replace "at" with "along".

Line 103 delete "the" at the end of the line. Line 106 replace "it" with "Korsfjord" to reduce the use of generic "it" and be specific to the subject under description. Line 108 remove full-stop and replace with and to join sentences for better flow. Line 109 correct "witch" to "which". Line 118 replace "mix" with "mixes". Line 119 insert second bracket ")" after "(NCC". Line 121 change "wind" to "winds" and insert "the water in" after "circulation of". Line 122 replace "that follows" with "following". Line 124 suggestion to add "forcing" after "wind". Line 125 replace "," with "and inputs of". Line 125-126 between "seasonal time scales," and "during spring-summer" replace text with "salinity drives stratification". Line 126 add "the water column is" between "and" and "more". Page 5

Line 129 specify what "the temperature" refers to, i.e., a certain depth or depth range

in the water column or certain location in the fjord. Line 131 delete "the" start line with "Water". Line 132 following "oxygen" insert "to the area". Line 132-133 delete "the fjords enhance their" and insert "is enhanced in the fojrd" between "production" and "which". Line 133 replace "enables them to host" with "supports". Line 137 replace "decisive" with "dominant controls of" and delete "for the". Line 140 write "nutrient-rich" and add "on from" after "follows". Line 142 insert "upper" before "water column", i.e., assuming that the sub-surface/ deeper waters remain nutrient rich? Line 143 insert 2 "," either side of "with its adjacent waters". Line 152 replace "It" with "The ship". Line 153 replace "." after "(Fig. 1)" with "," and delete "It".

Line 6

Line 161 replace "in" with "during". Line 166 add SOCAT reference: Bakker, D. C. E., Response: all the above correction have been implemented in the revised text.

Line 192 instead of capital letter use lower case "s" for "station". Line 200 clarify the estimated pH values – estimated from what source? CO2SYS?

Response: this is clarified in the revised ms, by modifying the sentence to "*These sensor data* were used to assess the uncertainty in our pH values estimated as described in the next section."

2.4 Methods

Line 206-207 insert "an" before "empirical" and define the parameters used to derive the empirical relationship.

Response: this is clarified in the revised ms, by modifying the end of the sentence to "...sea surface salinity (SSS) was determined from empirical relationship with SST."

Line 217 replace "Second" with "Secondly". Line 217 clarify how TA is derived from SSS(sst), see general comments. Response: see response to the general comments.

3. Results and discussion
3.1 Correlations and validations
Line 236 insert "the" before "data".
Line 237-239 Is sea ice present in the region? see general comments.
Response: see response to the general comments.

Line 274 insert "through the calculations" between "propagated" and "to".

Response: in the revised text (page 11), the sentence have been modified to "*These two error* estimates were combined (as the square root of sum of squares) to determine the total error in our estimates, which were found to be  $\pm 0.01$  and  $\pm 0.1$  for pH and  $\Omega Ar$ , respectively."

Line 239 clarify where the "runoff" originates. Line 240 insert

"degree of" between "high" and "scatter". Line 244 reverse order of "indeed is" to "is indeed". Line 247 delete "a" after "As" and re-write sentence following "verification" as "that the RF SST dataset is spatially representative,". Line 249 replace "in" with "across". Line 256 replace "statics" with "statistics". Line 257 replace "functions" with "acts" and insert "an" between "as" and "indicator". Line 259 add a space to separate "with" and "values". Line 260 add commas to read "data ,i.e., pH(sss)". Line 265 insert "closely" between "

Ar(sst)" and "reproduce" and delete "very well". Line 269 replace "show" with "shown".

3.2 Spatiotemporal variations

Line 281 replace "collapsed" with "condensed". Line 284 delete "the" following "after". Line 286 replace "outweighs" with "counteracts". Line 287 replace "begins" with "begin" and also specify what processes are being referred to. Line 288 insert "the" after September replace "SSTs" with "SST". Line 289 re-phrase sentence between "mixing," and "as mentioned" to read "which enables deep, carbon-rich coastal water to penetrate the surface layer,". Line 290 replace "as" between "and" and "reflected" with "is" and delete "the" before "increasing". Perhaps specify which DIC values are increasing by the autumnal mixing, i.e., sea surface, surface layer, upper ocean: : : Line 293 replace "drives up the " with "enriches". Line 294 insert "the" before "concentration", delete "the" between "of" and "carbonate" and add an s to read "ions". Line 296 rephrase to read "due to inputs of run-off" and replace "reinforces" with "enhances". Line 297 replace "over" with "exhausted".

Response: all the above corrections have been implemented in the revised text.

Line 298 replace "The" with "However," and delete

", on the other hand,".

Response: we find this sentence clear in the original text.

Line 300 replace "mismatch" with "decoupling of". Line 307 correct "embody" to "embodied".

3.3 Controls of seasonal variability and trends

Line 310 replace "arranged the data into" with "computed". Page 11 Line 315 re-write the reference to "Lauvset et al. (2015)". Line 316-317 replace "are shown on Fig. 4 (left panels) where it can be seen" with "show" – removes the double Fig. reference in the sentence. Line 326 replace "letting one of the drivers" with "varying them independently".

Line 327 delete "to vary" and replace "drivers" with "controls" – removing double word use in the same sentence. Line 329 replace "for" with "of" and correct "driver" to "drivers". Line 331 correct "induce" to "induces". Line 332 insert "other" between "the" and "hand". Line 333 correct "induce" to "induces". Line 334-335 re-write to "We therefore conclude that variations in DIC, followed by TA, are the most important drivers for changes in

Ar(sst)." Line 337 replace "have" with "has" and delete "In

terms of processes" so that the next line starts with "This means". Line 338 correct to "carbon-rich". Line 339 insert "inputs" after "runoff". Line 343 replace "changes" to "change".

Response: all the above corrections have been implemented in the revised text.

The first use of nDIC – how is defined and how are the normalised values determined, perhaps add reference, e.g., Friis et al. (2003).

Response: in the revised text (page 13), this part of the sentence has been modified to "...both for SST and DIC normalized to the mean salinity (nDIC) according to Friis et al. (2003)"

3.4 Inference of OA parameters from VOS underway data
Line 353 the use of DICS is not specifically defined in the text and is an additional
term to nDIC – do they both refer to salinity normalised DIC determined in the same
way? Please clarify to be sure.

Response: please see the general comments regarding "DICS".

Line 362 remove ", while the CO2 system is fully determined only occasionally," as I don't think it adds anything it would reduce the length of the sentence.

Response: we chose to keep "while the  $CO_2$  system is fully determined only occasionally" in the text in order to emphasize that occasional determination of the full CO2-system is necessary for the method to work.

Line 355 remove the space after (2010). Line 359 reference to equations 4 and 5 might not be correct, i.e., should it be equations 3 and 4. Please check and amend if necessary.

Line 368 delete the spare "and" after "Fig. 5" and add "to" between "conform" and "tight". Line 371 replace extra "," with ";" after "10.354" in equation 5.

Line 372 check for extra spaces and uses of "," and/ or ";" in separation of terms and consistency with equation 5. Line 377 delete "and" before "thus" and replace "min- imise" with "minimizing". Line 380 replace "with" by "to". Line 391 change to "carbonrich". Line 392 rearrange the line to read "surface water also reflects the properties of the deeper water". Line 394 replace "can" with "could" and insert "to occur" after "expected". Line 399 replace "development" with "CO2 concentrations" and replace "follows" with "follow". Line 400 add "driving oceanic CO2 uptake" after "atmosphere". 4. Summary and concluding remarks

Line 402 insert "worth" after "four years" and replace "less" with "sporadic". Line 403 delete "frequent". Line 413-414 use the same number of decimal places for quoted

numbers. Line 415 insert "the" before "phytoplankton" and replace ", which" with "that" and delete "levels". Line 417 replace "brings up" with "allows"; correct to "carbonrich"; insert "mix into" between "to" and "the". Line 418-419 replace "and reinforces the decrease in pH" with "reduces" and delete ", which continuous throughout fall". Line 430-431 finish sentence after "carbonate system".

Line 435 replace "development" with "concentrations" and remove "f" from "fCO2", i.e., referring to concentrations of CO2 rather than the concentrations of the fugacity of CO2 that wouldn't be quite correct. Line 436 finish the sentence after "atmosphere".

5. Acknowledgements

Line 438 replace "supports" with "support" and replace "by" with "from". Line 440 insert "the" after "and help of".

Tables

Table 1

Page 20 Line 598 replace "about" with "of". Line 599 insert "which" after "for".

Page 21 Line 603 replace "(continue)" with "(continued)".

Table 3

Page 22 nDIC – "DIC normalised to the mean salinity"; please include somewhere in the main body of the text what value was determined for the mean salinity and how. Figure texts

Page 23 Line 620 replace "which" with "where" and replace "tick" with "thick". Line 635 check "ln(nfCO2ts)" – is the "n" correct and currently it doesn't match with the X-axis labelling in Figure 5 (B).

Figures

Figure 2 (A) Y-axis can be expanded to open up the visual regression trend, i.e., range in SSS of 25-35. Confusion of the colour bar underneath is too close to the X-axis and on first inspection looks like it is referring to SST values. Add "month" as the unit to clarify the colour bar and move it closer to panel (A) if possible. Figure (3) 4 time series panels are presented but only 2 parameters mentioned in the Figure text and all colour bars are unlabelled. Add parameter labels and units (where relevant) to each of the coloured panels.

Response: all the above corrections have been implemented in the revised text.

[revised manuscript text omitted]
 fiords in western                                                                                                                                                                                                                                                                                                                                                                                                                                                                                                                                                                                                                                                                                                                                                                                                                                                                                                                                                                                                                                                                                                                                                                                                                                                                                                                                                                                                                                           | Formatted: Foot color: Text 1                                                                                                                                                                                                                                        |
|                                                                                                    | I ming the knowledge gap on or 1 (and generally the carbon eyere) in Ijords in western                                                                                                                                                                                                                                                                                                                                                                                                                                                                                                                                                                                                                                                                                                                                                                                                                                                                                                                                                                                                                                                                                                                                                                                                                                                                                                                                                                                                                                     |                                                                                                                                                                                                                                                                      |
| 70                                                                                                 | Norway has not been described previously. Filling this knowledge gap Norwagian fiords is                                                                                                                                                                                                                                                                                                                                                                                                                                                                                                                                                                                                                                                                                                                                                                                                                                                                                                                                                                                                                                                                                                                                                                                                                                                                                                                                                                                                                                          | Formatted: Font color: Text 1                                                                                                                                                                                                                                        |
| 78                                                                                                 | Norway has not been described previously. Filling this knowledge gap Norwegian fjords is                                                                                                                                                                                                                                                                                                                                                                                                                                                                                                                                                                                                                                                                                                                                                                                                                                                                                                                                                                                                                                                                                                                                                                                                                                                                                                                                                                                                                                          | Formatted: Font color: Text 1 Formatted: Font color: Text 1                                                                                                                                                                                                          |
| 78
79                                                                                           | Norway has not been described previously. Filling this knowledge gap Norwegian fjords is
important because the fjordsthese areas are important spawning grounds for different fish                                                                                                                                                                                                                                                                                                                                                                                                                                                                                                                                                                                                                                                                                                                                                                                                                                                                                                                                                                                                                                                                                                                                                                                                                                                                                                                                             | Formatted: Font color: Text 1                                                                                                                |
| 78
79
80                                                                                     | 
[revised manuscript text omitted]
 itthe fjord branches into the smaller   | F   | formatted: Font color: Text 1 |  |
| 114 | and shallower fjords Lysefjord and Fanafjord. To, and to the southwest it connects with the     | F   | formatted: Font color: Text 1 |  |
| 115 | open coast through the Selbjørnsfjord, witch which has a sill depth of 180 m at Selbjørn. To    | F   | formatted: Font color: Text 1 |  |
| 116 | the south it connects to the Hardangerfjord through the 25 km long and 300 m deep strait        |     |                               |  |
| 117 | Langenuen.                                                                                      |     |                               |  |
| 118 | The Hardangerfjord is a 179 km long fjord ranking as the fourth longest fjord in the world. It  |     |                               |  |
| 119 | stretches from the coastal open ocean in the southwest to the mountainous interior of Norway.   |     |                               |  |
| 120 | Our study includes the southern parts of the fjord. This is bounded by the larger islands Stord |     |                               |  |
| 121 | and Tysnesøya in the north, the Haugaland peninsula in the south, and the smaller islands       |     |                               |  |
| 122 | Fjellbergøya and Halsnøya on the south/east side. This part of the fjord is over 300 m deep in  |     |                               |  |
| 123 | its basin (around 59.76N; 5.55E) and connects with the smaller fjords Ålfjord and Bjoafjord     |     |                               |  |
| 124 | in the south.                                                                                   |     |                               |  |
| 125 | In the fjord system run-off from land mixmixes with salty water originating from the            | F   | formatted: Font color: Text 1 |  |
| 126 | northward flowing Norwegian Coastal Current (NCC, to produce a typically salinity               | F   | Formatted: Font color: Text 1 |  |

present the mean distribution across the different fjords (Korsfjord-Langenuen-

[revised manuscript text omitted]

| 252 | pH and $\Omega_{Ar}$ values based on TA(sss) and fCO 2 will be denoted as pH(sss) and $\Omega_{Ar}$ (sss),                                                                                                                                                                                                                                                                                                                                                                                                                                                                                                                                                                                                                                                                                                                                                                                                                                                                                                                                                                                                                                                                                                                                                                                                                                                                                                                                                                                                                                                                                                                                                                                                                                                                                                                                                                                                                                                                                                                                                                                                         |          |
|-----|-------------------------------------------------------------------------------------------------------------------------------------------------------------------------------------------------------------------------------------------------------------------------------------------------------------------------------------------------------------------------------------------------------------------------------------------------------------------------------------------------------------------------------------------------------------------------------------------------------------------------------------------------------------------------------------------------------------------------------------------------------------------------------------------------------------------------------------------------------------------------------------------------------------------------------------------------------------------------------------------------------------------------------------------------------------------------------------------------------------------------------------------------------------------------------------------------------------------------------------------------------------------------------------------------------------------------------------------------------------------------------------------------------------------------------------------------------------------------------------------------------------------------------------------------------------------------------------------------------------------------------------------------------------------------------------------------------------------------------------------------------------------------------------------------------------------------------------------------------------------------------------------------------------------------------------------------------------------------------------------------------------------------------------------------------------------------------------------------------------------------------|----------|
| 253 | whereas values that are either measured or computed from measured TA and DIC will be                                                                                                                                                                                                                                                                                                                                                                                                                                                                                                                                                                                                                                                                                                                                                                                                                                                                                                                                                                                                                                                                                                                                                                                                                                                                                                                                                                                                                                                                                                                                                                                                                                                                                                                                                                                                                                                                                                                                                                                                                                          |          |
| 254 | denoted as simply pH and $\Omega_{Ar}$ . nDIC denotes the DIC values normalized to constant salinity                                                                                                                                                                                                                                                                                                                                                                                                                                                                                                                                                                                                                                                                                                                                                                                                                                                                                                                                                                                                                                                                                                                                                                                                                                                                                                                                                                                                                                                                                                                                                                                                                                                                                                                                                                                                                                                                                                                                                                                                                   |          |
| 255 | (the mean value) according to Friis et al. (2003) with freshwater end member DIC                                                                                                                                                                                                                                                                                                                                                                                                                                                                                                                                                                                                                                                                                                                                                                                                                                                                                                                                                                                                                                                                                                                                                                                                                                                                                                                                                                                                                                                                                                                                                                                                                                                                                                                                                                                                                                                                                                                                                                                                                                              |          |
| 256 | concentration of 1039 µmol kg -1 inferred from the cruise data. An overview of the symbols                                                                                                                                                                                                                                                                                                                                                                                                                                                                                                                                                                                                                                                                                                                                                                                                                                                                                                                                                                                                                                                                                                                                                                                                                                                                                                                                                                                                                                                                                                                                                                                                                                                                                                                                                                                                                                                                                                                                                                                                                         |          |
| 257 | used for estimated and derived quantities used in this study is given in Table 2.                                                                                                                                                                                                                                                                                                                                                                                                                                                                                                                                                                                                                                                                                                                                                                                                                                                                                                                                                                                                                                                                                                                                                                                                                                                                                                                                                                                                                                                                                                                                                                                                                                                                                                                                                                                                                                                                                                                                                                                                                                             |          |
| 258 | 3. Results and discussion                                                                                                                                                                                                                                                                                                                                                                                                                                                                                                                                                                                                                                                                                                                                                                                                                                                                                                                                                                                                                                                                                                                                                                                                                                                                                                                                                                                                                                                                                                                                                                                                                                                                                                                                                                                                                                                                                                                                                                                                                                                                                                     |          |
| 259 | 3.1 Correlations and validations                                                                                                                                                                                                                                                                                                                                                                                                                                                                                                                                                                                                                                                                                                                                                                                                                                                                                                                                                                                                                                                                                                                                                                                                                                                                                                                                                                                                                                                                                                                                                                                                                                                                                                                                                                                                                                                                                                                                                                                                                                                                                              |          |
| 260 | In this section we present the regression equations identified in this study in addition to                                                                                                                                                                                                                                                                                                                                                                                                                                                                                                                                                                                                                                                                                                                                                                                                                                                                                                                                                                                                                                                                                                                                                                                                                                                                                                                                                                                                                                                                                                                                                                                                                                                                                                                                                                                                                                                                                                                                                                                                                                   |          |
| 261 | validating the various estimation procedures used by comparing the estimated values with                                                                                                                                                                                                                                                                                                                                                                                                                                                                                                                                                                                                                                                                                                                                                                                                                                                                                                                                                                                                                                                                                                                                                                                                                                                                                                                                                                                                                                                                                                                                                                                                                                                                                                                                                                                                                                                                                                                                                                                                                                      |          |
| 262 | those measured/computed. The results of these comparisons are summarized in Table 3. For                                                                                                                                                                                                                                                                                                                                                                                                                                                                                                                                                                                                                                                                                                                                                                                                                                                                                                                                                                                                                                                                                                                                                                                                                                                                                                                                                                                                                                                                                                                                                                                                                                                                                                                                                                                                                                                                                                                                                                                                                                      |          |
| 263 | each comparison, the coefficient of determination (R 2 ) and the significance level ( p -value) are                                                                                                                                                                                                                                                                                                                                                                                                                                                                                                                                                                                                                                                                                                                                                                                                                                                                                                                                                                                                                                                                                                                                                                                                                                                                                                                                                                                                                                                                                                                                                                                                                                                                                                                                                                                                                                                                                                                                                                                                         |          |
| 264 | used as metrics for the goodness of the correlation while the associated root-mean-square                                                                                                                                                                                                                                                                                                                                                                                                                                                                                                                                                                                                                                                                                                                                                                                                                                                                                                                                                                                                                                                                                                                                                                                                                                                                                                                                                                                                                                                                                                                                                                                                                                                                                                                                                                                                                                                                                                                                                                                                                                     |          |
| 265 | error (rmse) is benchmarked against the maximum target uncertainties developed by the                                                                                                                                                                                                                                                                                                                                                                                                                                                                                                                                                                                                                                                                                                                                                                                                                                                                                                                                                                                                                                                                                                                                                                                                                                                                                                                                                                                                                                                                                                                                                                                                                                                                                                                                                                                                                                                                                                                                                                                                                                         |          |
| 266 | Global Ocean Acidification Network (GOA-ON) and the California Current Acidification                                                                                                                                                                                                                                                                                                                                                                                                                                                                                                                                                                                                                                                                                                                                                                                                                                                                                                                                                                                                                                                                                                                                                                                                                                                                                                                                                                                                                                                                                                                                                                                                                                                                                                                                                                                                                                                                                                                                                                                                                                          |          |
| 267 | Network (C-CAN) of ±0.2 for <math>\Omega_{Ar}</math> (McLaughlin et al., 2015), which corresponds to maximum                                                                                                                                                                                                                                                                                                                                                                                                                                                                                                                                                                                                                                                                                                                                                                                                                                                                                                                                                                                                                                                                                                                                                                                                                                                                                                                                                                                                                                                                                                                                                                                                                                                                                                                                                                                                                                                                                                                                                                                                           |          |
| 268 | uncertainties of $\pm 0.02$ , $\pm 1.25$ or $\pm 1.8$ in pH, SST, or SSS, respectively.                                                                                                                                                                                                                                                                                                                                                                                                                                                                                                                                                                                                                                                                                                                                                                                                                                                                                                                                                                                                                                                                                                                                                                                                                                                                                                                                                                                                                                                                                                                                                                                                                                                                                                                                                                                                                                                                                                                                                                                                                                       |          |
| 269 | The regional SST-SSS relationship obtained from the RF dataset is given by Eq. 1 and is                                                                                                                                                                                                                                                                                                                                                                                                                                                                                                                                                                                                                                                                                                                                                                                                                                                                                                                                                                                                                                                                                                                                                                                                                                                                                                                                                                                                                                                                                                                                                                                                                                                                                                                                                                                                                                                                                                                                                                                                                                       |          |
| 270 | depicted in Fig. 2a (filled symbols). Despite a clear covariation between SST and SSS, there                                                                                                                                                                                                                                                                                                                                                                                                                                                                                                                                                                                                                                                                                                                                                                                                                                                                                                                                                                                                                                                                                                                                                                                                                                                                                                                                                                                                                                                                                                                                                                                                                                                                                                                                                                                                                                                                                                                                                                                                                                  |          |
| 271 | is a lot of scatter in the data and the statistics of the regression equation is not particularly                                                                                                                                                                                                                                                                                                                                                                                                                                                                                                                                                                                                                                                                                                                                                                                                                                                                                                                                                                                                                                                                                                                                                                                                                                                                                                                                                                                                                                                                                                                                                                                                                                                                                                                                                                                                                                                                                                                                                                                                                             |          |
| 272 | strong (Eq. 1). The observed correlation most probably arises from the annual cycles; during                                                                                                                                                                                                                                                                                                                                                                                                                                                                                                                                                                                                                                                                                                                                                                                                                                                                                                                                                                                                                                                                                                                                                                                                                                                                                                                                                                                                                                                                                                                                                                                                                                                                                                                                                                                                                                                                                                                                                                                                                                  |          |
| 273 | summer the study area embodies warm water diluted by runoff, whereas during winter the                                                                                                                                                                                                                                                                                                                                                                                                                                                                                                                                                                                                                                                                                                                                                                                                                                                                                                                                                                                                                                                                                                                                                                                                                                                                                                                                                                                                                                                                                                                                                                                                                                                                                                                                                                                                                                                                                                                                                                                                                                        |          |
| 274 | surface water is colder and saltier due to little or no runoff. The magnitude of these annual                                                                                                                                                                                                                                                                                                                                                                                                                                                                                                                                                                                                                                                                                                                                                                                                                                                                                                                                                                                                                                                                                                                                                                                                                                                                                                                                                                                                                                                                                                                                                                                                                                                                                                                                                                                                                                                                                                                                                                                                                                 |          |
| 275 | variations varies with time and space and this is reflected by the high degree of scatter in the Formatted : Font color: Text 1                                                                                                                                                                                                                                                                                                                                                                                                                                                                                                                                                                                                                                                                                                                                                                                                                                                                                                                                                                                                                                                                                                                                                                                                                                                                                                                                                                                                                                                                                                                                                                                                                                                                                                                                                                                                                                                                                                                                                                                 |          |
| 276 | relationship. Consequently, the identified regression model is able to explain only 27% of the                                                                                                                                                                                                                                                                                                                                                                                                                                                                                                                                                                                                                                                                                                                                                                                                                                                                                                                                                                                                                                                                                                                                                                                                                                                                                                                                                                                                                                                                                                                                                                                                                                                                                                                                                                                                                                                                                                                                                                                                                                |          |
| 277 | salinity variations. Nonetheless, the independent station and sensor data (dots, squares, and                                                                                                                                                                                                                                                                                                                                                                                                                                                                                                                                                                                                                                                                                                                                                                                                                                                                                                                                                                                                                                                                                                                                                                                                                                                                                                                                                                                                                                                                                                                                                                                                                                                                                                                                                                                                                                                                                                                                                                                                                                 |          |
| 278 | stars), which have been acquired from the whole study area, fits well in different seasons, falls,                                                                                                                                                                                                                                                                                                                                                                                                                                                                                                                                                                                                                                                                                                                                                                                                                                                                                                                                                                                                                                                                                                                                                                                                                                                                                                                                                                                                                                                                                                                                                                                                                                                                                                                                                                                                                                                                                                                                                                                                                            |          |
| 279 | into a pattern around the relationship described by Eq.1 with an RMSa root-squared-mean                                                                                                                                                                                                                                                                                                                                                                                                                                                                                                                                                                                                                                                                                                                                                                                                                                                                                                                                                                                                                                                                                                                                                                                                                                                                                                                                                                                                                                                                                                                                                                                                                                                                                                                                                                                                                                                                                                                                                                                                                                       | 5        |
| 280 | error, of 0.81 psu. Thus, these data confirm from here on we assume that Eq. 1 indeed is                                                                                                                                                                                                                                                                                                                                                                                                                                                                                                                                                                                                                                                                                                                                                                                                                                                                                                                                                                                                                                                                                                                                                                                                                                                                                                                                                                                                                                                                                                                                                                                                                                                                                                                                                                                                                                                                                                                                                                                                                                      | 5        |
| 281 | representative forable to estimate the seasonal SSS variations across the whole study region.                                                                                                                                                                                                                                                                                                                                                                                                                                                                                                                                                                                                                                                                                                                                                                                                                                                                                                                                                                                                                                                                                                                                                                                                                                                                                                                                                                                                                                                                                                                                                                                                                                                                                                                                                                                                                                                                                                                                                                                                                                 |          |
| 282 | To verify this we have compared the monthly averages of RE_SSS data with values obtained                                                                                                                                                                                                                                                                                                                                                                                                                                                                                                                                                                                                                                                                                                                                                                                                                                                                                                                                                                                                                                                                                                                                                                                                                                                                                                                                                                                                                                                                                                                                                                                                                                                                                                                                                                                                                                                                                                                                                                                                                                      |          |
| 202 | Formatted: | $\dashv$ |
| 283 | using Eq.1 and monthly KF_SS1. As shown in the last row of Table 3, the estimated values                                                                                                                                                                                                                                                                                                                                                                                                                                                                                                                                                                                                                                                                                                                                                                                                                                                                                                                                                                                                                                                                                                                                                                                                                                                                                                                                                                                                                                                                                                                                                                                                                                                                                                                                                                                                                                                                                                                                                                                                                                      |          |

| 284        | were significantly correlated with the monthly RF_SSS ( $R^2=0.65$ and p=0.002) and the

---

## Author Response (AR2)

Dear Editor/ Dear Mario,

Thanks for the final comments to our revised manuscript, which requested minor revisions. Please find attached the revised manuscript in which I have addressed all of your comments except the following (which I believe is based on misunderstanding): "L289-292 "However, using a multi-parameter linear regression with SST and SSS as independent parameters improved the regression statistics considerably ($R^2$=0.90; n=23; rmse=13.0 µmolkg-1) compared to a linear regression with only SSS ($R^2$=0.67; n=23; rmse=24.0 µmolkg-1)." Isn't this an artefact as SSS is derived from SST?".

I would like to bring to your attention that the TA-SSS/SST relationship (Eq. 2) is based on regression between TA and measured SSS AND SST. Thus, the including the SST as regression parameter improves the regression because SST adds more independent information about the variations. Therefore, I chose not to address the above comment in the revised manuscript.

I would like to take the opportunity to thank you for handling the manuscript and helping us improve it, we appreciate your constructive comments/suggestions and encouragement.

Best Regards,
Abdir